# L-OPA1 regulates mitoflash biogenesis independently from membrane fusion

Manon Rosselin[1], Jaime Santo-Domingo[2], Flavien Bermont[1], Marta Giacomello[3] & Nicolas Demaurex[1,*] ID

## Abstract

Mitochondrial flashes mediated by optic atrophy 1 (OPA1) fusion protein are bioenergetic responses to stochastic drops in mitochondrial membrane potential ($\Delta\psi_m$) whose origin is unclear. Using structurally distinct genetically encoded pH-sensitive probes, we confirm that flashes are matrix alkalinization transients, thereby establishing the pH nature of these events, which we renamed "mitopHlashes". Probes located in cristae or intermembrane space as verified by electron microscopy do not report pH changes during $\Delta\psi_m$ drops or respiratory chain inhibition. *Opa1* ablation does not alter $\Delta\psi_m$ fluctuations but drastically decreases the efficiency of mitopHlash/$\Delta\psi_m$ coupling, which is restored by re-expressing fusion-deficient OPA1[K301A] and preserved in cells lacking the outer-membrane fusion proteins MFN1/2 or the OPA1 proteases OMA1 and YME1L, indicating that mitochondrial membrane fusion and OPA1 proteolytic processing are dispensable. pH/$\Delta\psi_m$ uncoupling occurs early during staurosporine-induced apoptosis and is mitigated by OPA1 overexpression, suggesting that OPA1 maintains mitopHlash competence during stress conditions. We propose that OPA1 stabilizes respiratory chain supercomplexes in a conformation that enables respiring mitochondria to compensate a drop in $\Delta\psi_m$ by an explosive matrix pH flash.

**Keywords** bioenergetics; membrane fusion; mitoflash; OPA1
**Subject Categories** Autophagy & Cell Death; Membrane & Intracellular Transport

## Introduction

Proton transfer across the inner membrane of mitochondria (IMM) couples the oxidation of carbohydrates and fat to the synthesis of adenosine triphosphate (ATP) that cells use as an energy source. During oxidative phosphorylation, the flow of electrons along four respiratory complexes of the electron transport chain (CI to CIV) is coupled to the flux of protons from the mitochondrial matrix to the cristae [1–3], creating a proton-motive force ($\Delta p$) across the IMM composed of an electrical component, the mitochondrial membrane potential ($\Delta\psi_m$ ~180 mV, negative inside), and a chemical component, the transmembrane pH gradient ($\Delta pH_m$ ~0.8, alkaline inside) [4,5]. The $\Delta p$ drives protons back to the matrix across the $F_0F_1$ ATP synthase or complex V (CV), thereby generating ATP [1,6]. In the cristae, the complexes of the electron transport chain assemble into functional quaternary structures termed respiratory chain supercomplexes (RCS) with enhanced electron transport and respiration efficiency [7–9].

The assembly and stability of RCS depends on the mitochondrial-shaping protein optic atrophy 1 (OPA1), an inner membrane protein which forms oligomeric complexes to sustain cristae architecture [10,11]. OPA1 is a dynamin-related GTPase whose best characterized function is to coordinate the fusion of mitochondrial inner membranes [12–14], while the fusion of mitochondrial outer membranes is mediated by mitofusin proteins MFN1 and MFN2 [15–17]. The *OPA1* gene is alternatively spliced at exons 4, 4a, and 5b leading to the expression of eight variants that differ in their ability to trigger membrane fusion and in their resistance to apoptosis [18–20]. These variants can be constitutively cleaved by the IMM peptidase OMA1 and the *i*-AAA protease YME1L at sites located in exon 5 and 5b, respectively [21–23]. OPA1 cleavage leads to the production of short, soluble forms of OPA1 (S-OPA1) that interact with uncleaved forms of OPA1 (L-OPA1) at cristae junctions and with subunits of the mitochondrial contact site (MICOS) complex involved in the maintenance of cristae structure [24–26]. Recent findings also suggest that S-OPA1 is involved in IMM fission [27].

Respiring mitochondria exhibit spontaneous bursts of chemical and electrical activity, a phenomenon known as mitochondrial flash (mitoflash) which we recently showed to be dependent on OPA1 [28]. Mitoflashes detected with the circularly permuted yellow fluorescent protein targeted to mitochondria (mito-cpYFP) were initially interpreted as bursts of superoxide production [29] but subsequent studies revealed that cpYFP reports pH changes and is insensitive to superoxide [30–33]. Mitoflashes are readily detected with mito-sypHer, a cpYFP-derived probe with enhanced pH sensitivity, and correspond to reversible alkalinizations of the matrix of ~0.4 pH units that coincide in time and in space with transient $\Delta\psi_m$ depolarization events [28,29]. These opposite changes in the electrical and chemical components of $\Delta p$ compensate each other thermodynamically, and clamping $\Delta\psi_m$ at different potentials evokes compensatory matrix pH (pH$_{mito}$) changes. This indicates that

1   Department of Cell Physiology and Metabolism, University of Geneva, Geneva, Switzerland
2   Nestlé Institute of Health Sciences SA, EPFL Innovation Park, Lausanne, Switzerland
3   Department of Biology, University of Padova, Padova, Italy
    *Corresponding author. Tel:+41 22 379 5399; Fax: +41 22 379 5338; E-mail: nicolas.demaurex@unige.ch

transient matrix alkalinizations preserve the ability of mitochondria to convert energy during drops in $\Delta\psi_m$ [28].

Mitoflashes were reported in a wide range of cells *in vitro* and in intact heart and skeletal muscle [29,34] and shown to require a functional respiratory chain [28,29], but their mechanistic basis remains unclear. The $\Delta\psi_m$ drops that trigger mitoflashes are thought to reflect transient opening of the mitochondrial permeability transition pore (mPTP), but the impact of mPTP modulators on mitoflashes varies depending on the cell or tissue tested [29,34,35]. Earlier studies proposed that $\Delta\psi_m$ fluctuations reflect local $Ca^{2+}$ or $Na^+$ elevations [36–39], opening of mPTP [36,38,40,41], coupling of $\Delta\psi_m$ to the ATP synthase [42], or switching between active and inactive states of oxidative phosphorylation [43]. Hyperosmotic or oxidative stress and ROS-dependent apoptosis are consistently associated with increased mitoflash frequency, defining mitoflashes as quantitative bioindicators of cellular dysfunctions [35,44–46]. In *Caenorhabditis elegans,* mitoflash frequency correlates with life span and worms with low mitoflash activity live significantly longer than those with high mitoflash activity, highlighting the physiological relevance of these events [47].

In this study, we sought to clarify the mechanism of mitoflash generation by tracking the transfer of protons across the IMM. Using different fluorescent pH sensors targeted to mitochondrial compartments, we consistently observed matrix alkalinization transients, establishing the pH nature of the flashes, which we renamed "mitopHlashes". However, we failed to detect pH changes in the cristae or in the intermembrane space (IMS) during drops in $\Delta\psi_m$. We further show that OPA1 is not required for stochastic $\Delta\psi_m$ fluctuations but allows efficient generation of mitopHlashes, that L-OPA1 isoforms can restore flashing independently of mitochondrial fusion, and that loss of OPA1-mediated mitopHlash/$\Delta\psi_m$ coupling is an early marker of apoptosis.

## Results

### Mitoflashes are transient matrix alkalinization events

The mitoflash probe cpYFP is pH-sensitive and does not respond to superoxide [31,33] but the notion that mitoflashes reflect bursts of superoxide still persists [29,47–49]. To confirm the pH nature of mitoflashes, we used two pH-sensitive proteins structurally distinct from cpYFP: pHred and the super-ecliptic pHluorin (spHluorin). pHred is a ratiometric pH indicator derived from the red fluorescent protein mKeima [50], and spHluorin is a non-ratiometric pH probe derived from GFP [51]. We targeted these probes to the mitochondrial matrix by fusing pHred to the signal sequence of the cytochrome c oxidase subunit VIII (Cox8) and spHluorin to the C-terminus of the matrix soluble protein peptidase (MPP) or to a subunit of the CV facing the matrix (CVɣ) (Rieger B *et al*, in preparation) (Fig EV1A). The mitochondrial localization of the three pH sensors was verified by confocal microscopy (Fig 1A–C). All probes co-localized with the resident mitochondrial protein Hsp60 with Manders and Pearson coefficient around 80% (Fig EV1B–D). pH titration curves were obtained by exposing transfected HeLa cells to nigericin and monensin to equilibrate the matrix pH with the external pH (Fig 1D). Increasing the pH from 5 to 8 decreased the pHred intensity ratio ($F_{561}/F_{405}$) by more than 10-fold with the expected pKa of 6.7 [50], while the fluorescence of MPP- and CVɣ-spHluorin

increased with a pKa of 7.2. The average resting pH values reported with Cox8-pHred, MPP- and CVɣ-spHluorin at 37°C were 7.75 ± 0.42, 7.90 ± 0.32 and 7.85 ± 0.34, respectively (mean ± SD), consistent with previous matrix pH measurements [52–55]. Addition of antimycin A, an inhibitor of the respiratory complex III, evoked fluorescence changes corresponding to an acidification (Figs 1F and EV2A and B) in 96% of cells expressing Cox8-pHred ($n = 29$) and in 70% of cells expressing MPP-spHluorin ($n = 57$) or CVɣ-spHluorin ($n = 47$), indicating that the three probes report dynamic changes in matrix pH. We then tested whether mitoflashes could be recorded with these matrix pH sensors. Antiparallel fluorescence transients occurring spontaneously and randomly in space were readily detected with Cox8-pHred, reporting matrix pH flashes phenocopying the mitoflashes (Fig 1G and Movie EV1). Reversible fluorescence increases corresponding to transient alkalinization transients were detected in single mitochondria of cells expressing spHluorin fused to CVɣ and MPP. Simultaneous recording of pH and $\Delta\psi_m$ using these probes and tetramethyl rhodamine methyl ester (TMRM) confirmed that the matrix pH flashes always occurred concomitantly with drops in $\Delta\psi_m$ without any lag observed between the initiation phase of these two signals (Fig 1H and I, and Movie EV2). The decay phases of pH and $\Delta\psi_m$ transients did not always mirror each other as mitochondrial potential sometimes exhibited a delayed recovery. The kinetics pH flashes reported by the new probes and by the previously validated matrix pH probe mito-sypHer were comparable, the probes reporting events of similar duration (Cox8-pHred: 7.68 ± 1.79 s, CVɣ-spHluorin: 8.2 ± 1.98 s, MPP-spHluorin: 7.35 ± 1.26 s, and mito-sypHer: 8.6 ± 0.6 s) and time to peak (Cox8-pHred: 3.06 ± 1.12 s, CVɣ-spHluorin: 1.91 ± 0.83 s, MPP-spHluorin: 1.70 ± 0.44 s, and mito-sypHer: 1.63 ± 0.6 s). The flashing area measured with all the sensors was comparable and drastically increased upon enforced mitochondrial fusion with a dominant negative mutant of dynamin-related protein 1 (DRP1[K38A], data not shown), consistent with our previous findings [28]. Thus, pH flash events sharing similar spatial and temporal properties are detected in the matrix of mitochondria with structurally distinct pH sensors. This establishes pH as the main contributor of mitochondrial flashes, which we therefore renamed "mitopHlashes".

### MitopHlashes are not detectable in the IMS and in cristae

We [28] and others [29,31] previously postulated that mitopHlashes reflect a burst in proton pumping by the respiratory chain. To validate this hypothesis, we used pH-sensing proteins addressed to the cristae space and the IMS to detect acidifying bursts in these compartments. Two cristae pH probes based on spHluorin fused to the C-terminal of the subunit 8a of complex IV (CIV8) and to the subunit e of complex V (CVe) were kindly provided by Karin Busch [56], and we fused ratiometric pHluorin (rpHluorin [51]) to the mitochondrial localization sequence of Smac (Smac-rpHluorin) to measure IMS pH (Fig EV1). The probes exhibited a mitochondrial pattern when transfected cells were imaged by confocal microscopy (Fig 2A–C) and co-localized with the endogenous marker Hsp60 (Fig EV1B–D). pH calibrations revealed that spHluorin fluorescence increased upon alkalinization with a pKa of 7.2, while the Smac-rpHluorin $F_{488}/F_{405}$ ratio decreased with a pKa of 7.1 (Fig 2D). The resting pH values reported by these sensors were more acidic than those obtained with the probes targeted to the matrix, consistent with their predicted

    

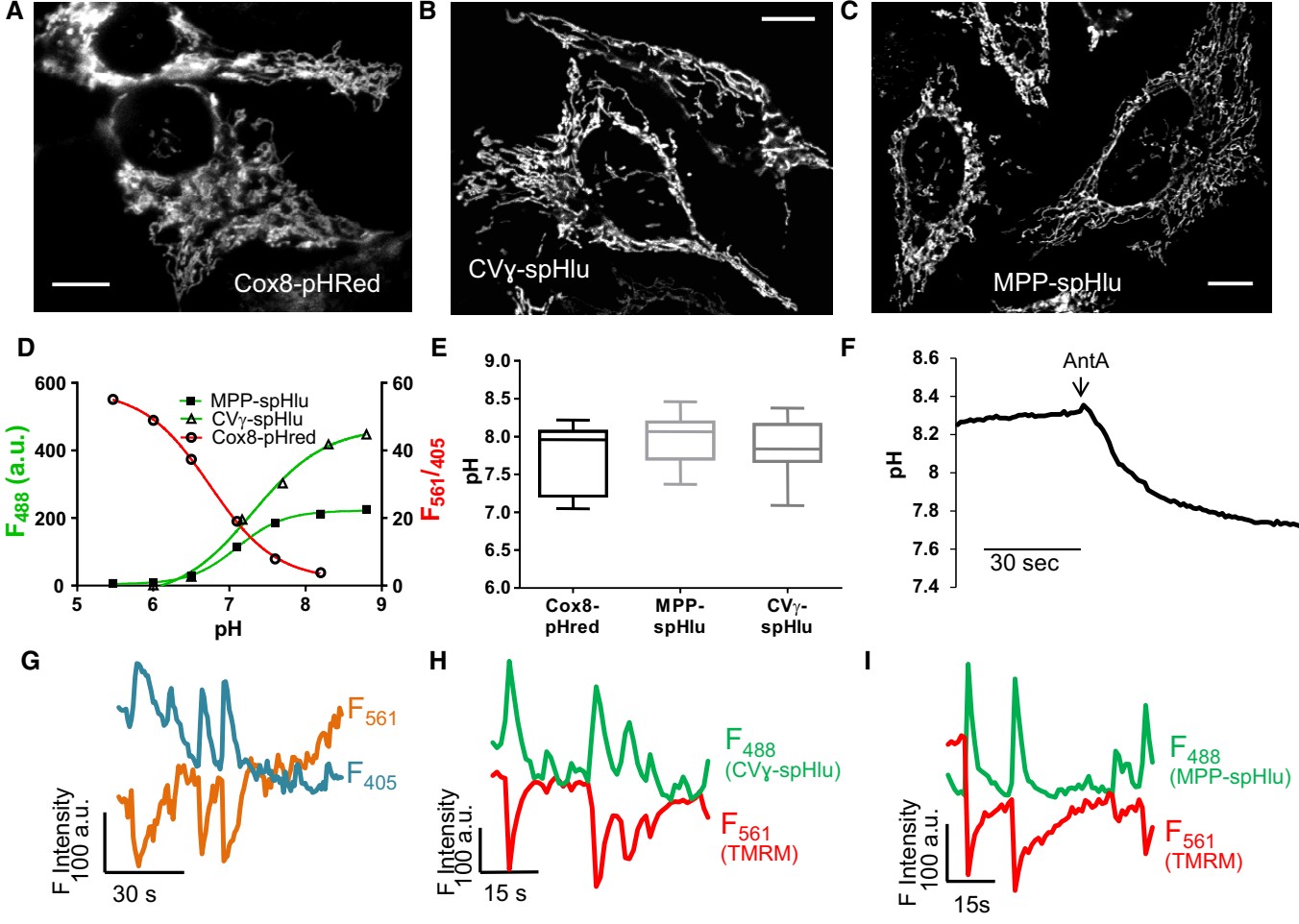

**Figure 1. Matrix alkalinization "flashes" detected with structurally different pH sensors.**

A–C    Confocal images of (A) Cox8-pHred, (B) CVɣ-spHluorin, and (C) MPP-spHluorin in HeLa cells. Scale bars: 10 µm.

D    *In situ* pH calibration of pHred ($F_{561}/F_{405}$ excitation ratio), CVɣ-, and MPP-spHluorins ($\lambda_{ex}$: 488 nm).

E    Average resting pH reported by the three probes. Box and whisker plots show the 25th–75th percentiles, median, and maximal values from three independent experiments.

F    Representative curve showing change in Cox8-pHred $F_{561}/F_{405}$ ratio upon addition of antimycin A (AntA, 5 µM), corresponding to an acidification of the matrix. Similar responses were obtained with CVɣ- and MPP-spHluorins (Fig EV3A and B).

G    Antiparallel changes in Cox8-pHred fluorescence at $\lambda_{ex}$ of 405 and 561 nm corresponding to matrix alkalinization transients.

H, I    Simultaneous recordings of TMRM fluorescence and matrix pH with (H) CVɣ- or (I) MPP-spHluorin reporting alkalinization events during drops in $\Delta\psi_m$.

Data information: See also Movies EV1 and EV2.

localization in the cristae and IMS (Fig 2E). However, the intensiometric cristae probes reported values more alkaline than the ratiometric IMS probe (Smac-rpHluorin: 6.98 ± 0.34, CVe-spHluorin: 7.49 ± 0.22 and CIV8-spHluorin: 7.51 ± 0.21, average ± SD) or than an earlier study reporting a resting cristae pH of 7.2 [56]. We then checked the effects of acute inhibition of respiratory complexes. Inhibition of complex III with antimycin, expected to alkalinize the IMS and cristae, did not alter the smac-rpHluorin ratio (Fig 2F) or the fluorescence of the two cristae spHluorins (Fig EV2C and D). Inhibition of complex V with oligomycin was also without effect (Fig EV2E). However, when cells were switched to a galactose-rich or to a low glucose media, the resting pH$_{mito}$ values reported with CVe-spHluorin decreased to 7.25 ± 0.06 and 6.97 ± 0.06, respectively (Fig EV2F), indicating that the probe reports chronic changes in the pH of cristae. Given the large amplitude of matrix mitopHlashes, we expected to

detect variations in IMS and cristae pH in mitochondria exhibiting stochastic depolarization transients. Contrary to this prediction, time-resolved measurements did not reveal any transient pH change during $\Delta\psi_m$ drops (Fig 2G–I and Movie EV3), and mitopHlashes remained undetectable in cells grown in galactose to boost aerobic metabolism (Fig EV2G). To enhance the detection of potential mitoflash activity in the cristae or the IMS, we next expressed the DRP1 mutant DRP1$^{K38A}$, which extends the connectivity of mitochondria and increases the flashing areas (Fig EV3) [28,57,58]. Although DRP1$^{K38A}$ expression led to mitochondrial depolarization transients that could spread over the entire mitochondrial network, pH changes remained undetectable in the cristae or the IMS during $\Delta\psi_m$ drops (Fig EV3). To verify that the expression of proteins fused to respiratory chain complexes does not inhibit flashing activity, we co-expressed the matrix-targeted pH sensor mito-sypHer together with the cristae and IMS probes. The same

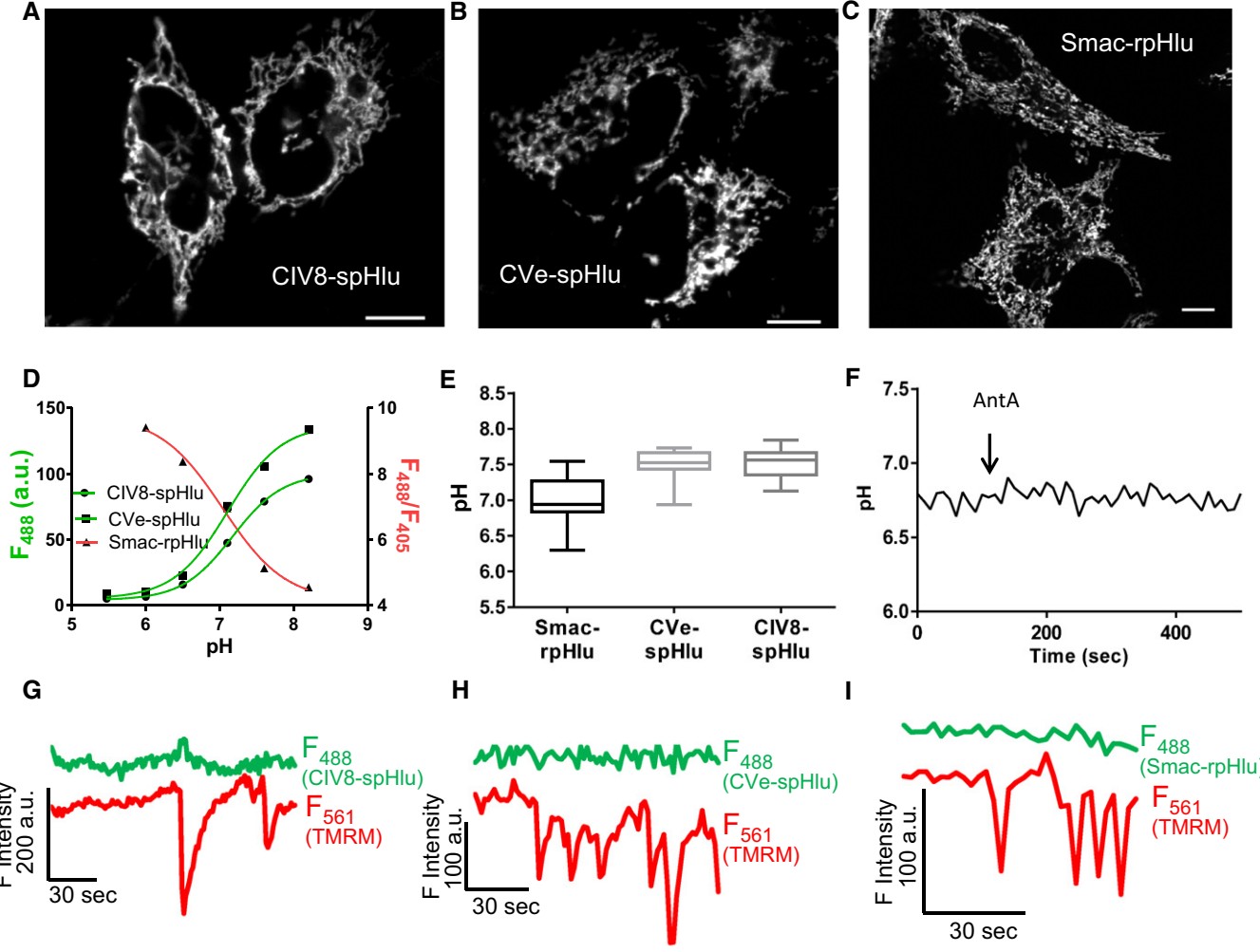

**Figure 2.  pH events are not detectable in the IMS or in the cristae.**

A–C   Confocal images of (A) CIV8-spHluorin, (B) CVe-spHluorin, and (C) Smac-rpHluorin in HeLa cells. Scale bars: 10 μm.

D   *In situ* pH calibration of Smac-rpHluorin ($F_{488}/F_{405}$ excitation ratio), CIV8- and CVe-spHluorins ($\lambda_{ex}$: 488 nm).

E   Average resting pH reported by the three probes. Box and whisker plots show the $25^{th}$–$75^{th}$ percentiles, median, and maximal values from three independent experiments.

F   Representative curve showing change in Smac-rpHluorin fluorescence evoked by 5 μM antimycin A (AntA). Similar responses were obtained with CIV8- and CVe-spHluorins (Fig EV3C and D).

G–I   Concomitant recordings of $\Delta\psi_m$ (TMRM) and pH measured with (G) CIV8-, (H) CVe-spHluorin, or (I) Smac-rpHluorin reporting no detectable pH changes during drops in $\Delta\psi_m$.

Data information: See also Movie EV3.

---

frequency of mitopHlashes was recorded in cells expressing cristae or IMS fusion proteins, indicating that flashing activity is maintained in these conditions (Fig EV2H). We then considered that the failure to detect acute pH changes in the IMS and cristae could reflect improper localization of the genetically encoded pH sensors, and performed immunoelectron microscopy with anti-GFP antibodies to precisely assess their sub-mitochondrial localization. MPP-spHluorin immunoreactivity was preferentially detected in the matrix, while CIV8-spHluorin gold particles were mainly detected in cristae, confirming their correct targeting (Fig 3). MitopHlashes are therefore not associated with detectable flashing activity using pH probes verifiably located in IMS and cristae, suggesting that protons are not transferred across the mitochondrial inner membrane during flashes.

## OPA1 modulates mitopHlash/$\Delta\psi_m$ coupling independently of mitochondrial fusion

We previously showed that OPA1 is necessary for flashing activity [28]. To determine the contribution of OPA1 in this process, we performed concomitant measurements of mitopHlashes and $\Delta\psi_m$ in wild-type (WT) and $Opa1^{-/-}$ mouse embryonic fibroblasts (MEF) expressing mito-sypHer. In WT cells, transient drops in TMRM fluorescence strictly coincided with mitopHlashes and 97% of depolarization transients were concomitant with mitopHlashes (Fig 4A and E). *Opa1* ablation caused a drastic loss in mitopHlashes while $\Delta\psi_m$ drops persisted unabatedly (Fig 4B), at a frequency similar to WT cells (0.9 ± 0.04 vs. 1.2 ± 0.43 depolarization/cell/min for WT and

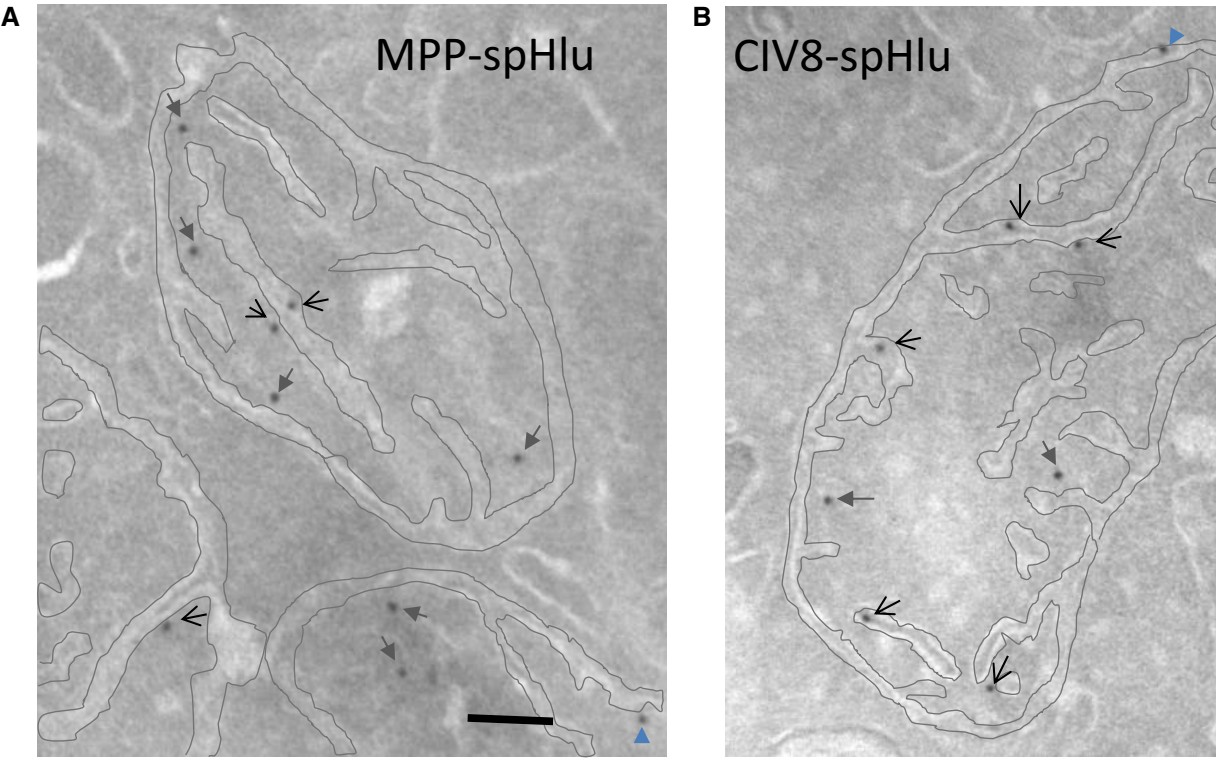

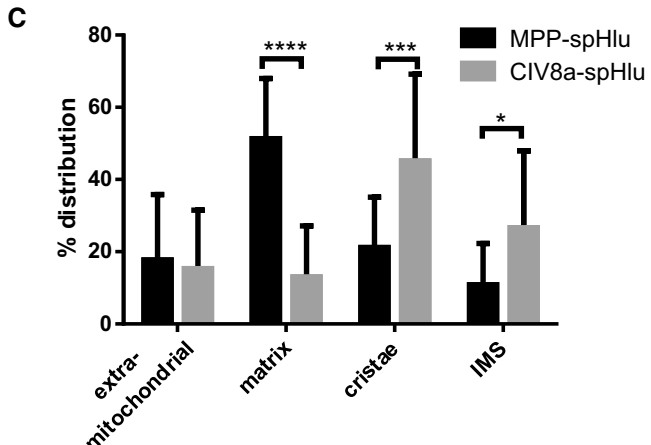

**Figure 3.  Localization of CIV8- and MPP-spHluorins in mitochondrial compartments.**

A, B    Electron microscopy images showing the localization of gold-labeled anti-GFP antibodies in HeLa cells expressing (A) MPP- and (B) CIV8-spHluorins. The IMM and OMM of mitochondria are outlined in gray to better visualize the different mitochondrial sub-compartments. Black opened arrows, gray filled arrows, and blue arrowheads correspond to a cristae, matrix, and IMS localization, respectively. Scale bars: 100 nm.

C    Distribution of gold particles in the different compartments. Values are means ± SD. 81 mitochondria with 269 gold particles were analyzed for MPP-spHluorin and 47 mitochondria with 169 gold particles for CIV8-spHluorin. Two-way ANOVA, *P = 0.0211, ***P = 0.0005, ****P < 0.0001.

$Opa1^{-/-}$, Fig 4D). As a result, the coupling between depolarization and pH transients dropped from 97% in WT cells to 4.4% in $Opa1^{-/-}$ cells (Fig 4E). Re-expression of WT OPA1 (variant 1) partially restored the coupling between mitopHlashes and $\Delta\psi_m$ drops to 34% (Fig 4C and E).

As the best characterized function of OPA1 is to mediate IMM fusion, we next investigated whether mitochondrial fusion is necessary to couple $\Delta\psi_m$ drops and mitopHlashes [28]. The role of outer

mitochondrial membrane (OMM) fusion was explored using cells lacking the two mitofusins ($Mfn1/2^{-/-}$) that coordinate OMM fusion and the role of IMM fusion by re-expressing either WT OPA1 or the fusion-deficient mutant OPA1$^{K301A}$ in $Opa1^{-/-}$ cells [59]. Monitoring the fusion of individual mitochondria by photoactivation of mito-PA-GFP confirmed that OPA1, but not OPA1$^{K301A}$, restored mitochondrial fusion in $Opa1^{-/-}$ cells (Fig 5A and B). The frequencies of $\Delta\psi_m$ fluctuations were comparable in all conditions except in

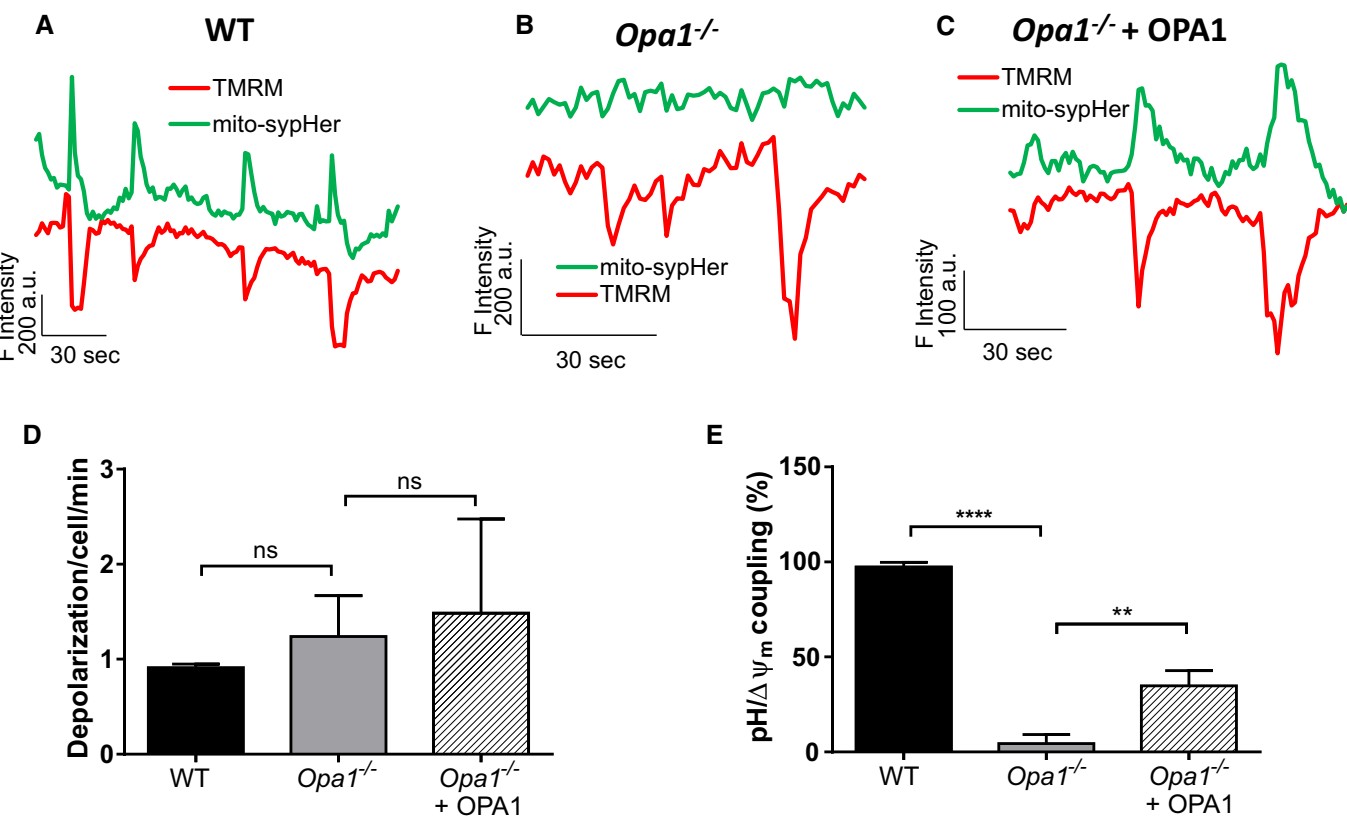

**Figure 4. OPA1 modulates mitopHlash/Δψ$_m$ coupling.**

A–C    Concurrent recordings of Δψ$_m$ (TMRM) and matrix pH (mito-sypHer, λ$_{ex}$: 488 nm) in (A) WT, (B) *Opa1$^{-/-}$*, or (C) *Opa1$^{-/-}$* cells re-expressing OPA1.

D, E    Depolarization frequency (D) with n = 41, 25, and 42 cells recorded in WT, *Opa1$^{-/-}$*, and *Opa1$^{-/-}$* cells re-expressing OPA1, respectively, and coupling between mitopHlashes and Δψ$_m$ (E) (number of mitopHlashes/number of depolarizations) in the indicated cells with n = 65, 59, and 84 depolarization events recorded in 41, 25, and 42 WT, *Opa1$^{-/-}$* and *Opa1$^{-/-}$* cells re-expressing OPA1, respectively. Values are means ± SD of three independent experiments. One-way ANOVA with multiple comparisons, **$P$ = 0.0013, ****$P$ < 0.0001, ns: not significant.

*Opa1$^{-/-}$* cells expressing OPA1$^{K301A}$ which exhibited a 3.7-fold increase in TMRM spike frequency (Fig 5C), hinting to a stress condition. TMRM fluorescence images showing depolarizations of individual mitochondria in WT, *Opa1$^{-/-}$* cells transfected with a control plasmid, OPA1, or OPA1$^{K301A}$ are shown in Fig EV4. MitopHlashes were preserved in *Mfn1/2$^{-/-}$* cells and were equally restored by re-expression of WT and OPA1$^{K301A}$ in *Opa1$^{-/-}$* cells (53% ± 7.8 vs. 63% ± 27.5 of TMRM spikes associated with mitopHlashes, respectively, Figs 5D and EV4). These results demonstrate that mitofusins, OMM fusion, and the fusogenic activity of OPA1 are dispensable for mitopHlash/Δψ$_m$ coupling.

**Each long OPA1 variant can mediate mitopHlash/Δψ$_m$ coupling without proteolytic processing**

Fibroblasts express two splice variants of OPA1: variant 7 (V7), cleaved by the OMA1 and YME1L proteases at sites S1 and S2, respectively, and variant 1 (V1), cleaved only at S1 by OMA1 (Fig 6A) [21–23]. Processing of these two variants leads to the production of three short soluble forms of OPA1 (S-OPA1) that interact with the long, uncleaved forms of OPA1 (L-OPA1) to form oligomeric complexes at cristae junctions (Fig 1) [25,26]. To gain insight into the role of OPA1, we studied the individual contribution of the

two splice variants in Δψ$_m$ drops and mitopHlashes. We measured Δψ$_m$ and matrix pH in *Opa1$^{-/-}$* cells expressing mito-SypHer and either flag-tagged V1 or V7. Each variant restored pH/Δψ$_m$ coupling when expressed in *Opa1$^{-/-}$* cells, without altering the frequency of Δψ$_m$ fluctuations (Fig 6B and C), the variant 7 being slightly, but not significantly, more potent in restoring mitopHlashes (Fig 6B). To investigate the contribution of S-OPA1, we used cells derived from *Oma1$^{-/-}$*, *Yme1L$^{-/-}$*, or double *Oma1$^{-/-}$Yme1L$^{-/-}$* knockout mice [27]. Simultaneous recordings of membrane potential and pH flashes showed that the ablation of *Oma1*, *Yme1L*, or both genes had no effect on Δψ$_m$ fluctuations and their coupling to mitopHlashes (Figs 6D and E, and EV4). This indicates that OPA1 processing and thereby S-OPA1 are dispensable for mitopHlash generation.

**mitopHlash/Δψ$_m$ coupling is lost early during STS-induced apoptosis**

During apoptosis, destabilization of OPA1 high-molecular-weight complexes leads to a conformational change of cristae [10,25,26], a process associated with cytosolic release of OPA1 and cytochrome c [25]. As the coupling between Δψ$_m$ drops and mitopHlashes is mediated by OPA1, we hypothesized that pH/Δψ$_m$ coupling would be lost in apoptotic cells. To test this possibility, we recorded

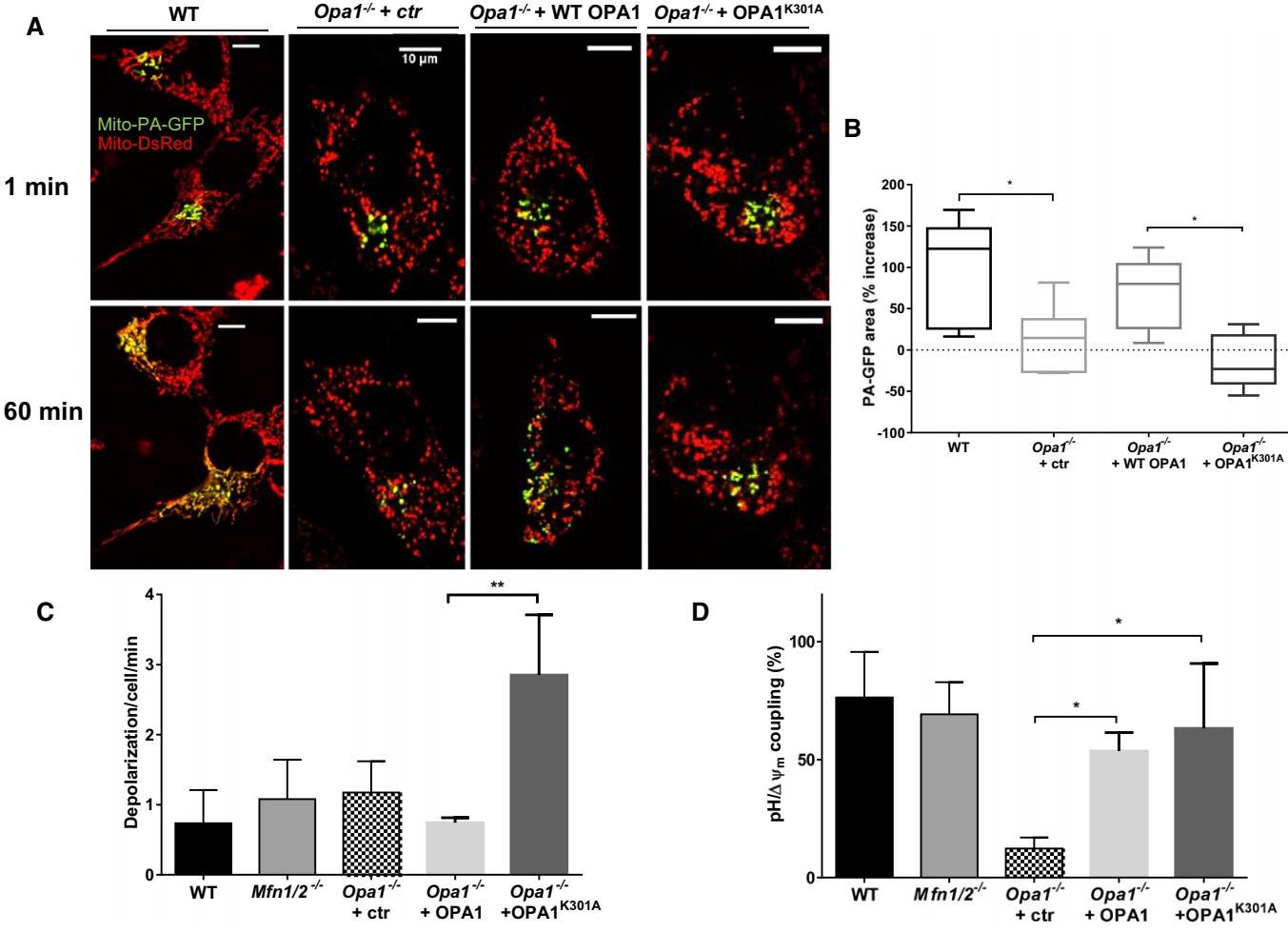

**Figure 5. Mitochondrial fusion is not required for mitopHlashes.**

A    Confocal images of matrix-targeted photoactivable GFP (mito-PA-GFP) and mito-DsRed co-expressed in WT, *Opa1*$^{-/-}$ cells transfected with control plasmid (ctr, pcDNA3), OPA1, or OPA1$^{K301A}$. Mitochondrial fusion was assessed by tracking the area of PA-GFP fluorescence 1 min (top) and 60 min (bottom) after photoactivation in the same cell. Scale bars: 10 μm.

B    Quantification of mitochondrial fusion as percentage of PA-GFP fluorescent area increase. Box and whisker plots show the 25$^{th}$–75$^{th}$ percentiles, median, and maximal values from three independent experiments (n = 7, 6, 6, and 7 cells for WT, *Opa1*$^{-/-}$ cells transfected with control plasmid (ctr, pcDNA3), OPA1, or OPA1$^{K301A}$, respectively). One-way ANOVA with multiple comparisons, *P = 0.0463 (WT vs. *Opa1*$^{-/-}$ + ctr), *P = 0.0329 (*Opa1*$^{-/-}$ + WT OPA1 vs. *Opa1*$^{-/-}$ + OPA1 $^{K301A}$).

C, D    Depolarization frequency (C) and mitopHlash/Δψ$_m$ coupling (D) in WT, *Mfn1/2*$^{-/-}$, and *Opa1*$^{-/-}$ cells and in *Opa1*$^{-/-}$ cells re-expressing OPA1 or the fusion-deficient mutant OPA1$^{K301A}$. Values are means ± SD of three independent experiments with n = 129, 79, 194, 167, and 177 depolarization events recorded in 30, 28, 54, 43, and 41 cells, respectively. One-way ANOVA with multiple comparisons, *P = 0.0407 (*Opa1*$^{-/-}$ + ctr vs. *Opa1*$^{-/-}$ + OPA1), *P = 0.0172 (*Opa1*$^{-/-}$ + ctr vs. *Opa1*$^{-/-}$ + OPA1$^{K301A}$), **P = 0.0019.

matrix pH and Δψ$_m$ fluctuations in cells exposed to staurosporine (STS, 1 μM). We observed a fivefold increase in depolarization frequency 2 h after STS addition compared to the DMSO control condition (Fig 7A). The increased Δψ$_m$ flickering was followed by a complete collapse of Δψ$_m$ in most cells after 3 h of STS exposure (data not shown). STS also triggered a significant uncoupling between Δψ$_m$ and pH and only 37% of Δψ$_m$ drops were associated with mitopHlashes 2 h after drug exposure compared to 92% in control conditions (Fig 7B). The STS-induced pH/Δψ$_m$ uncoupling was observed in cells depolarizing at both low and high frequency and was not associated with a change in matrix pH reported with mito-SypHer (average pH of 7.54 ± 0.09 and 7.44 ± 0.02 after 2 h of DMSO and STS treatment, respectively). To investigate whether

pH/Δψ$_m$ uncoupling is a general feature of apoptosis, we also recorded pH and Δψ$_m$ fluctuations over time in MEF cells exposed to 250 mM H$_2$O$_2$ or 100 μM etoposide (Fig EV5). These two death triggers had the same effect as STS, suggesting that the loss of coupling between Δψ$_m$ drops and mitopHlashes is a conserved event during apoptosis. Western blot and immunofluorescence analyses showed that pH/Δψ$_m$ uncoupling occurs before cytosolic release of OPA1 and cytochrome c (Fig 7C and D), implying that an increased depolarization frequency and a loss of pH/Δψ$_m$ coupling are early indicators of STS-induced apoptosis. As OPA1 overexpression was shown to delay the apoptotic process and cristae remodeling [25,60], we tested whether pH/Δψ$_m$ uncoupling could be mitigated by enforced OPA1 expression. Simultaneous

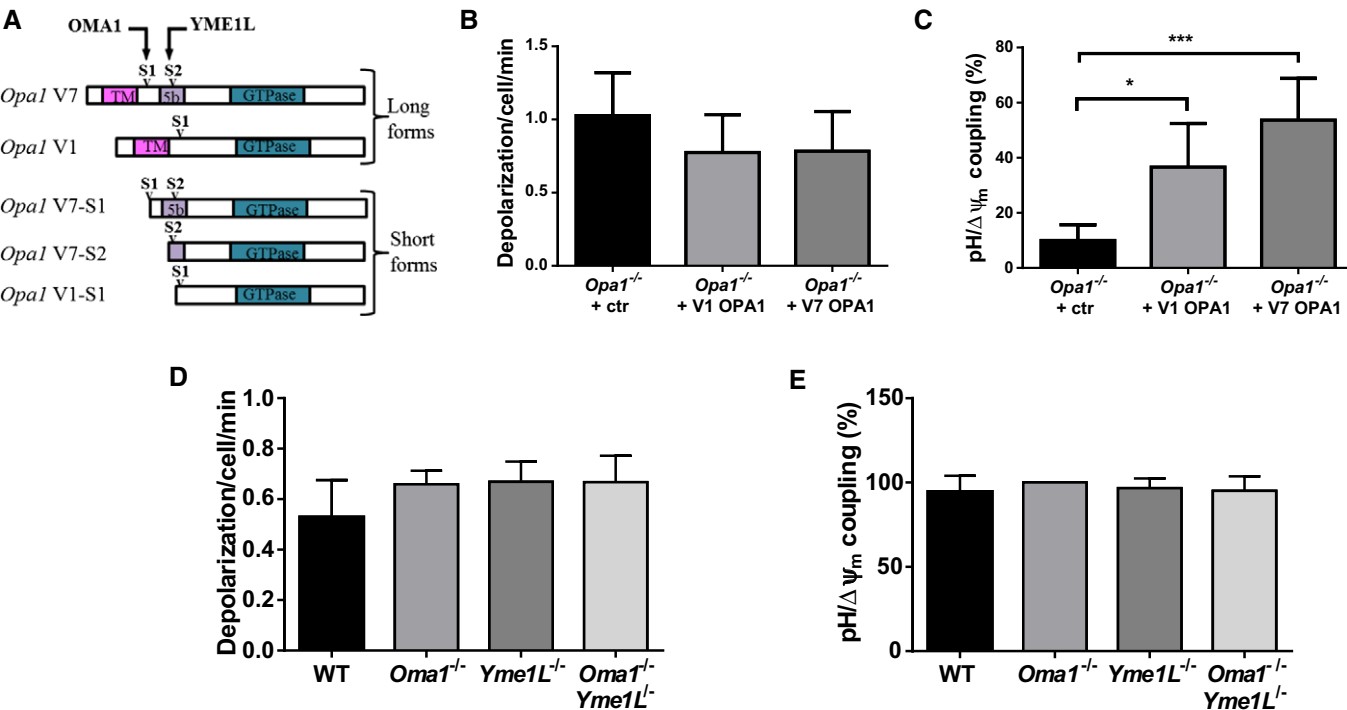

**Figure 6. Each long OPA1 variant can mediate mitopHlash/$\Delta\psi_m$ coupling without proteolytic processing.**

A       Schematic representation of the long and short forms of OPA1 derived from splice variants 1 (V1) and 7 (V7) and their corresponding short forms produced by proteolytic cleavage at sites S1 or S2 by OMA1 or YME1L, respectively.

B, C    Depolarization frequency (B) and mitopHlash/$\Delta\psi_m$ coupling (C) in $Opa1^{-/-}$ cells encoding pcDNA3 (ctr), re-expressing the long V1 or V7 OPA1 variant. *n* = 102, 198, and 93 depolarization events recorded in 21, 69, and 29 cells, respectively.

D, E    Depolarization frequency (D) and mitopHlash/$\Delta\psi_m$ coupling (E) in WT, $Oma1^{-/-}$, $Yme1L^{-/-}$, or $Oma1^{-/-}$ $Yme1L^{-/-}$ cells. *n* = 94, 100, 77, and 90 depolarization events recorded in 41, 40, 39, and 43 cells, respectively.

Data information: Values are means ± SD of three independent experiments. One-way ANOVA with multiple comparisons, *$P$ = 0.0152, ***$P$ = 0.0008.

recordings of $\Delta\psi_m$ and pH in MEF cells exposed to STS for 2 h revealed that OPA1 overexpression significantly reduces pH/$\Delta\psi_m$ uncoupling (Fig 7E). The fraction of cells displaying a low pH/$\Delta\psi_m$ coupling (0 to 25%) dropped from 33.69 to 9.3% when OPA1 was overexpressed while the fraction of highly coupled cells (75 to 100%) increased from 25 to 48.84%. These data indicate that the uncoupling of mitopHlashes from $\Delta\psi_m$ fluctuations is an early marker of apoptosis that is dependent on the levels of OPA1, suggesting that the cristae-shaping function of OPA1 contributes to mitopHlash/$\Delta\psi_m$ coupling.

## Discussion

In this study, we confirm that mitoflashes are matrix alkalinization events and provide new mechanistic insight on this bioenergetic phenomenon. We recorded prototypical mitoflashes with three structurally distinct matrix-targeted pH-sensitive proteins (pHred, spHluorin, and SypHer), thereby establishing the pH nature of these events, which we renamed "mitopHlashes". We then attempted to track the flow of protons across the mitochondrial respiratory membrane but failed to detect any pH flashing activity in cristae or the IMS. The resting pH reported by cristae probes was not modified by respiratory chain inhibitors and was more alkaline than previously reported

[56], despite cristae probe localization verified by electron microscopy. This raises the possibility that mitopHlashes are confined within the matrix and not associated with proton pumping by respiratory complexes as previously postulated. Alternatively, the association of respiratory complexes into supercomplexes to optimize ATP synthesis could direct the flux of protons into a specific circuit that bypasses the pH sensors fused to the CIV and the CV. MitopHlashes correlate with cellular stress and organism life span, and their underlying molecular mechanism is the object of intense research. We previously reported that *Opa1* ablation abrogates mitopHlash activity recorded with mito-SypHer and proposed that the OPA1-mediated formation of a fusion pore between adjacent mitochondria could equilibrate their potential, leading to compensatory mitopHlash responses [28]. We show here with combined $\Delta\psi_m$ and pH measurements that the frequency of $\Delta\psi_m$ fluctuations is unaffected in $Opa1^{-/-}$ cells, indicating that OPA1 is not necessary for the depolarization that triggers the mitopHlash response. Using TMRM spikes as indicators of activity, we could then detect residual mitopHlashes events in $Opa1^{-/-}$ cells, indicating that *Opa1* ablation severely impairs, but does not abrogate mitopHlash activity. OPA1 is therefore neither required for the generation of the electrical events nor for their conversion into a chemical response, but modulates the efficiency of pH/$\Delta\psi_m$ coupling. We also extend our previous findings linking the spatial spread of mitopHlashes to mitochondrial

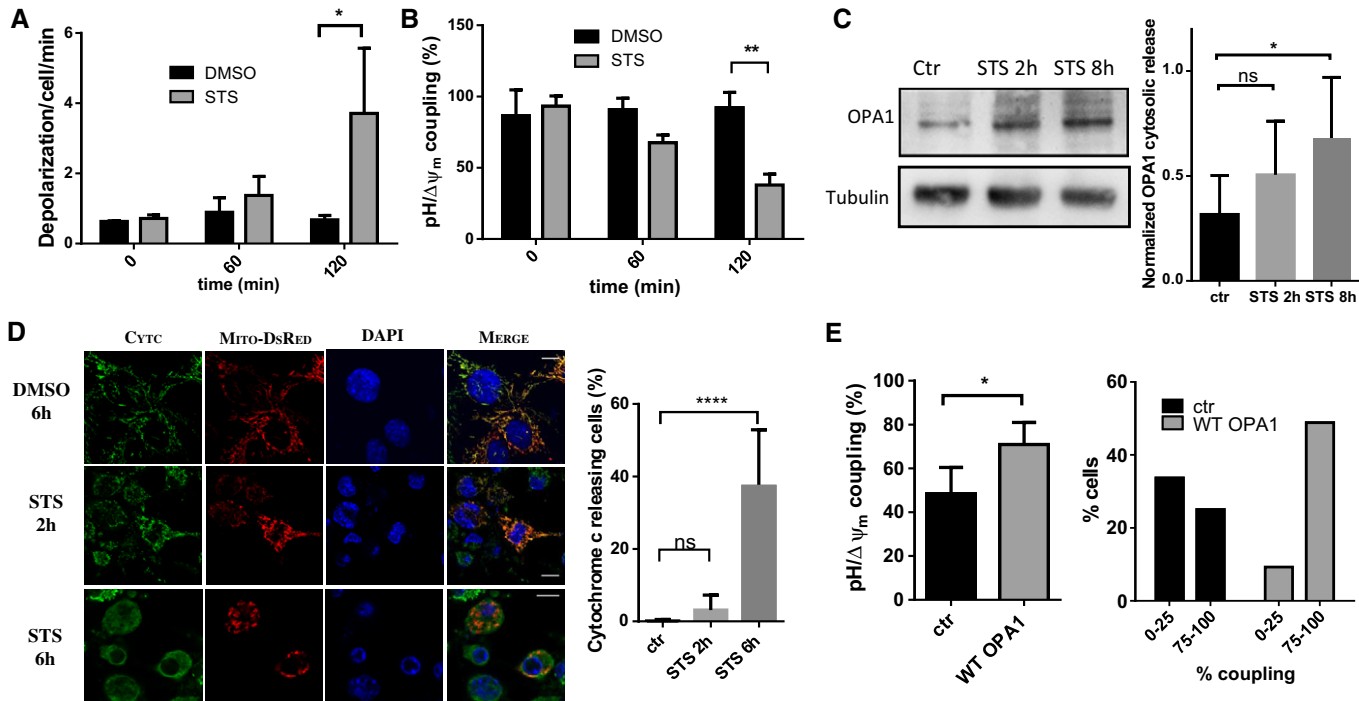

**Figure 7.  mitopHlash/Δψ_m uncoupling occurs early during apoptosis and is prevented by OPA1 overexpression.**

A, B   Depolarization frequency (A) and mitopHlash/Δψ_m coupling (B) in WT cells exposed to DMSO or 1 μM STS. Values are means ± SD of three independent experiments. Two-way ANOVA, *P = 0.0163, **P = 0.0085. n = 72, 99, 58 depolarization events recorded in 22, 27, and 20 cells for DMSO and 62, 75, 103 depolarization events recorded in 18, 22, and 24 cells for STS at 0, 60, and 120 min, respectively.

C   Western blots of OPA1 and tubulin performed on cytosolic fractions from WT cells exposed to DMSO or STS. The quantification of cytosolic OPA1 normalized to the amount of tubulin for each condition is shown. Values are means ± SD of three independent experiments, *P = 0.0332, paired t-test.

D   Confocal images showing cytochrome c immunoreactivity (green), mito-DsRed (red), and DAPI (blue) fluorescence 2 or 6 h after STS exposure. Note the mitochondrial localization of cytochrome c after 2 h of STS. Scale bars: 10 μm. Right panel shows the fraction of cells displaying cytoplasmic cytochrome c immunoreactivity. Values are means ± SD of three independent experiments, ****P < 0.0001, one-way ANOVA with multiple comparisons.

E   mitopHlash/Δψ_m coupling 2 h after STS exposure in WT MEF cells overexpressing OPA1 (variant 1) or control plasmid (ctr). Values are means ± SD of three independent experiments. Unpaired t-test with Welch's correction, *P = 0.0293. The fraction of cells with low pH/Δψ_m coupling (0–25%) and high pH/Δψ_m coupling (75–100%) is indicated. n = 422 and 312 depolarization events recorded in 57 and 44 cells for control (ctr) and OPA1 re-expression.

connectivity [28] by showing that mitopHlashes persist in *Mfn1/2*^−/− cells and are restored in *Opa1*^−/− cells by the re-expression of a fusion-deficient OPA1^K301A mutant. These observations show that MFN-mediated OMM fusion and OPA1-mediated IMM fusion are dispensable for pH/Δψ_m coupling.

Picard *et al* [61] recently reported that adjacent mitochondria can coordinate their metabolic state by forming intermitochondrial junctions characterized by parallel alignments of their cristae. This trans-mitochondrial communication could enhance bioenergetic efficiency across functional clusters of mitochondria during high energy demand and could provide the structural basis for rapid electrochemical events along connected mitochondria. In support for this, we previously reported that a high proportion (66%) of mitopHlashes bears adjacent mitochondria that coincidentally hyperpolarize [28], suggesting that mitopHlashes are associated with specific intermitochondrial contact sites. Correlative light-electron microscopy will be required to unravel the ultrastructure of flashing mitochondria and determine whether specific mitochondrial or cristae arrangements are associated with mitopHlash activity.

OPA1 is known to maintain cristae shape via the formation of oligomers which assemble at cristae junctions to staple cristae

membranes and tighten cristae necks [25,26]. In *Yme1L*^−/− *Oma1*^−/− cells that do not generate S-OPA1, cristae morphology remains unaltered [27], suggesting that L-OPA1 is sufficient to form oligomers that sustain cristae architecture. During STS and t-Bid-induced apoptosis, OPA1 oligomers are disrupted as cristae are remodeled [25,26]. We observed that the coupling between Δψ_m and mitopHlashes was significantly decreased upon addition of different apoptotic triggers, consistent with a role of OPA1 oligomers in maintaining a permissive cristae architecture. Our observations that mitopHlashes persist in cells lacking OPA1 proteolytic processing and that each long OPA1 variant individually reconstitutes mitopHlashes in *Opa1*^−/− cells indicate that the formation of S-OPA1 and L-OPA1 hetero-oligomers is not required for mitopHlashes. These data also show that S-OPA1 variants known to co-localize with the mitochondrial fission machinery [27] and to interact with subunits of the cristae-organizing MICOS complex [24] are dispensable for flashing.

During apoptosis, cristae remodeling is associated with the disassembly of RCS, whose formation and stability also depends on OPA1 [10]. RCS disassembly decreases respiratory efficiency and might impair the ability of mitochondria to respond to Δψ_m fluctuations. RCS disassembly could thus account for the loss of pH/Δψ_m coupling upon *Opa1* ablation and STS-induced apoptosis.

Conversely, OPA1 oligomerization during starvation leads to a tightening of cristae junctions and an increased ATP production by oxidative phosphorylation [60]. As mitopHlashes occur in immobile mitochondria [44] that are found at sites of high ATP demand [62,63], we speculate that OPA1 oligomers maintain flashing mitochondria in a high energetic state by enhancing RCS assembly, which in turn couple mitopHlashes to $\Delta\psi_m$ fluctuations.

In summary, we confirm that mitopHlashes are matrix-limited pH responses to $\Delta\psi_m$ drops and show that this metabolic pH/$\Delta\psi_m$ coupling disappears early during apoptosis and requires long forms of OPA1 but not mitochondrial fusion. One possible role of OPA1 during mitopHlash biogenesis could be the stabilization of RCS to allow explosive proton pumping by the respiratory chain to compensate drops in membrane potential.

# Materials and Methods

### Cell lines, transfection, and media

HeLa cells were cultured in modified Eagle's medium (MEM) + glutamax (Gibco, 41090-028). WT, $Opa1^{-/-}$, $Mfn1/2^{-/-}$, $Oma1^{-/-}$, $Yme1L^{-/-}$, and $Oma1^{-/-}Yme1L^{-/-}$ MEFs were grown in Dulbecco's modified Eagle's medium (DMEM) (Gibco, 22320-022). All media contained 10% fetal bovine serum (FBS), 1% penicillin and 1% streptomycin. For fluorescence imaging, cells were seeded and grown on 25-mm glass cover slips and transfected with 5 μl Lipofectamine 2000 (Invitrogen) and 2 or 6 μg DNA for expression of fluorescent or non-fluorescent proteins, respectively. During imaging, cells were incubated in a solution containing 20 mM N-2-hydroxyethylpiperazine-N′-2-ethanesulfonic acid (HEPES), 140 mM NaCl, 5 mM KCl, 1 mM MgCl₂, 2 mM CaCl₂, 10 mM glucose, pH set to 7.4 at 37°C. To assess the mitochondrial localization of pH sensors, 5 μM antimycin A (Sigma) or 5 μg/ml oligomycin (Calbiochem) was added to the imaging media. High glucose medium was replaced with galactose or low glucose medium when indicated using DMEM without glucose (Gibco, 11966-025) supplemented with 10 mM galactose (Sigma) or 5.5 mM glucose (sigma), 10% FBS, sodium pyruvate (Sigma, S-8636), 3.7 g/l sodium bicarbonate, 1 mM sodium pyruvate, 2 mM glutamine, 1% penicillin, 1% streptomycin, and 25 mM HEPES for 2 days.

### pH sensors used in the study

Mito-sypHer, Cox8-pHred, CVe-, and CIV8-spHluorins were already characterized [50,55,56], spHluorin being a derivate of GFP (F64L/S65T/S147D/N149Q/V163A/S175G/S202F/Q204T/A206T [51]). MPP- and CVγ-spHluorins were a kind gift from K. Busch (Rieger B *et al*, in preparation). The subunit e of the CV and MPP was fused with their C-termini to the N-terminus of spHluorin. Smac-rpHluorin was constructed by PCR amplification of the mitochondrial localization sequence of Smac (60 amino acids) using the primers 5′-GAGAGCTAGCATGGCGGCTCTGAAGAG-3′ (forward) and 5′-GGGG GGATCCTACTCCAAAGCCAATC-3′ (reverse) and the template vector pCi-SmacMTS-GFP-AviTag-Myc. It was then inserted into a pHluorin-N1 vector encoding the GFP-derived ratiometric pHluorin (S202H, E132D, S147E, N149L, N164I, K166Q, I167V, R168H, and L220F [51]). Smac was fused to the N-terminal extremity of rpHluorin.

### Co-localization between Hsp60 and pH sensors

HeLa cells expressing spHluorin, rpHluorin, or Cox8-pHred were fixed to coverslips with PBS containing 3% paraformaldehyde for 15 min and permeabilized with 0.5% Triton X-100 for 10 min. After blocking the coverslips in PBS-2% bovine serum albumin (BSA) Hsp60 was stained using a mouse monoclonal antibody (Abcam, ab46798 diluted 1:200, overnight at 4°C) and a secondary goat anti-mouse antibody conjugated to Alexa 594 (Molecular Probes, diluted 1:1,000, 1 h). Finally, coverslips were mounted in fluorescence mounting medium (Prolong, ThermoFisher) and analyzed with a Nikon A1R confocal microscope. The Pearson and Mander's coefficients were obtained using the Imaris software.

### pH calibration and determination of resting pH$_{mito}$

For pH titration, Cox8-pHred was alternately excited for 800 ms at 430 and 560 nm through ET430/24× and 560AF55 filters and imaged with a 645AF75 band pass filter (Omega Optical). spHluorin and rpHluorin were excited for 700 ms at 488 nm or at 405 and 488 nm, respectively, through 380AF15, and 480BP10 filters (Omega Optical) and imaged with a 535AF26 band pass filter. This was performed at 37°C on an Axio Observer microscope (Zeiss, Germany) equipped with a Lambda DG4 illumination system (Sutter Instrument Company, Novato, CA, USA). Images were typically acquired every 5 s. Image acquisition and analysis were performed with MetaFluor 6.3 software (Universal Imaging, West Chester, PA). Mitochondrial pH was calibrated using nigericin (5 μg/ml) and monensin (5 μm) in 125 mm KCl, 20 mm NaCl, 0.5 mm MgCl₂, 0.2 mm EGTA, and 20 mm N-methyl-D-glutamine (pH 9.5–10.0), Tris (pH 8.0, 9.0), HEPES (pH 7.0–7.5), or MES (pH 5.5–6.5). For each cell, a six-point calibration curve was fitted to a variable slope sigmoid equation using GraphPad Prism, and the resting pH$_{mito}$ was determined by reporting the initial fluorescence values to the calibration curves.

### Ratiometric changes of Cox8-pHred fluorescence and concomitant pH and potential measurements

Images were acquired on a Nikon a1r inverted confocal microscope with a ×60 objective (oil; CFI Plan APO 1.4 NA) and typically one image was acquired every 1 s. Rapid ratiometric changes of fluorescence measured with Cox8-pHred were recorded using two excitation wavelengths (405 and 561 nm) and one 624/40 emission filter. Time-resolved pH and potential imaging was performed at 37°C on cells transiently transfected with mito-sypHer or pHluorins and loaded with 4 or 20 nM TMRM for HeLa and MEF cells, respectively. Images were acquired using 488- and 561-nm excitations and 520/35 and 624/40 emission filters. The frequency of depolarizations and pH flashes was analyzed using ImageJ.

### Mitochondrial fusion activity

WT MEFs and $Opa1^{-/-}$ cells transfected with 6 μg pCXN2-OPA1 or OPA1$^{K301A}$ (gift from Prof. Yoshihiro Kubo) and 1 μg mito-PA-GFP and 0.5 μg mito-DsRed were imaged in DMEM media. Cells with low GFP fluorescence intensity were selected to avoid saturation of

the GFP emission upon photoactivation [62]. GFP and mito-DsRed were imaged concurrently on a Nikon A1R confocal microscope with a × 60 objective (oil; CFI Plan APO 1.4 NA) using 488 and 561 nm excitation and 520/35 and 624/40 emission filters. Three images were acquired (one image/second) before applying three stimulation pulses (500 ms, 405 nm laser 50 mW, 7% power) followed by live imaging. Loss of focus was minimized by using the Perfect Focus system (Nikon). ImageJ was used for data analysis of the area of PA-GFP. Fusion was quantified as the % of PA-GFP area increase after 60 min compared to the initial PA-GFP area measured 1 min post-photoactivation in the same cell.

### Electron microscopy

For immunoelectron microscopy, cells were fixed with 2% paraformaldehyde and 0.2% glutaraldehyde and processed for cryo-ultramicrotomy as described in [64]. Ultrathin frozen sections were prepared and incubated for immunolabeling as described in [65]. Rabbit polyclonal anti-GFP (Abcam, 1:200) were used as primary antibodies and goat anti-rabbit IgG gold as secondary antibodies (gold size, 15 nm). The analysis of gold particle localization was performed using the image analysis software Leica QWin Standard (Leica Imaging Systems Ltd.) and a Wacom graphic-pen tablet on electron micrographs at the final magnification ×93,500.

### Cytosolic release of OPA1 and cytochrome c

Mouse embryonic fibroblasts cells were treated with 1 μM STS for 2, 6, or 8 h or with DMSO for 6 or 8 h. Release of OPA1 in the cytoplasm was determined by Western blot after cytosolic fractionation. Cells were permeabilized with 8 mg/ml digitonin for 10 min and centrifuged at 10,000 g for 10 min. The supernatants were boiled at 99°C for 5 min prior to separation of proteins by SDS–PAGE. Proteins were transferred electrophoretically onto nitrocellulose membranes which were blocked for 1 h with TBS-0.05% Tween containing 5% non-fat dried milk at room temperature. Membranes were then incubated with mouse monoclonal anti-OPA1 (BD Transduction Laboratories, clone 18/OPA1 diluted 1:1,000) and mouse monoclonal anti-tubulin (Sigma, clone DM1A diluted 1:2,000). Peroxidase-conjugated anti-mouse sera were used as secondary antibodies (Bio-Rad, diluted 1:5,000). Detection was carried out using a chemiluminescence detection kit (Immobilon western, Millipore), and the bands were detected using Syngene PXi (Syngene) and the software Genesys V1.5.00. The amounts of cytosolic OPA1 and tubulin were quantified using ImageJ, and the level of OPA1 release was expressed as the ratio of OPA1/tubulin.

Cytochrome c release was assessed by immunofluorescence. MEF cells expressing mito-DsRed were fixed and permeabilized. After blocking the coverslips, cytochrome c was stained using a mouse monoclonal antibody (BD Pharmingen, clone 6H2.B4 diluted 1:200, overnight at 4°C) and a secondary goat anti-mouse antibody conjugated to Alexa 488 (Molecular Probes, diluted 1:1,000, 1 h). The nucleus was stained with 4′,6-diamidino-2-phenylindol (DAPI) for 30 min (Sigma). Finally, coverslips were mounted in fluorescence mounting medium (Prolong, ThermoFisher) and analyzed with a Nikon A1R confocal microscope. The number of cells releasing cytochrome c was determined using ImageJ.

### Statistical analysis

All statistical analyses were performed using GraphPad Prism version 6.05. Significance between three or more group sets was analyzed using one-way ANOVA. To analyze the statistical differences among flash frequencies and pH/$\Delta\psi_m$ coupling using STS and DMSO at different time points and among the distribution of MPP- and CIV8-spHluorins in the different cellular sublocalizations, a two-way ANOVA was performed. Statistical differences of pH/$\Delta\psi_m$ coupling in WT MEFs transfected with a control plasmid or OPA1 and of resting pH values in HeLa cells cultured in galactose or low glucose media were analyzed using an unpaired *t*-test with Welch's correction. The statistical test used to analyze OPA1 cytosolic release by Western blot is a ratio paired *t*-test. ns: non-significant, *$P < 0.05$, **$P < 0.01$, ***$P < 0.001$, ****$P < 0.0001$.

**Expanded View** for this article is available online.

### Acknowledgements

We thank Mr. Cyril Castelbou for expert technical assistance, Ms. Monica Bulla, Dr. Paula Nunes, Dr. Denis Martinvalet, and Prof. Claes Wolheim for their helpful comments. We also thank Prof. Jean-Claude Martinou for providing constructs encoding OPA1-flagged variants, Prof. Yoshihiro Kubo for pCXN2-OPA1 constructs, and Prof. Karin Busch for MPP-, CVγ-, CVe- and CIV8-spHluorins. *Oma1*$^{-/-}$, *Yme1L*$^{-/-}$, and *Oma1*$^{-/-}$ *Yme1L*$^{-/-}$ cells were a kind gift from Prof. Thomas Langer and *Mfn1/2*$^{-/-}$ MEF cells from Prof. David Chan. We also thank Pilar Ruga Fahy for her assistance in preparing electron microscopy samples. This work was supported by the Swiss National Science Foundation grant 31003A-149566.

### Author contributions

MR and ND designed experiments. MR and FB performed experiments and MR analyzed data. JS-D generated the pH sensor Smac-rpHluorin. MR, JS-D, MG, and ND drafted the article or revised it critically for important intellectual content.

### Conflict of interest

The authors declare that they have no conflict of interest.

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
