## [Review Process File · EMBO Reports]

Manuscript EMBO-2016-42931

L-OPA1 regulates mitoflash biogenesis independently from membrane fusion

Manon Rosselin, Jaime Santo-Domingo, Flavien Bermont, Marta Giacomello, and Nicolas Demaurex

Corresponding author: Nicolas Demaurex, University of Geneva

Review timeline:

Submission Date:	22 June 2016
Editorial Decision:	19 July 2016
Revision Received:	29 October 2016
Editorial Decision:	12 December 2016
Revision Received:	19 December 2016
Accepted:	21 December 2016

Editor: Barbara Pauly/Martina Rembold

Transaction Report:

1st Editorial Decision

19 July 2016

Thank you for the submission of your research manuscript to our journal. We have now received the full set of referee reports that is copied below.

As you will see, the referees acknowledge the potential interest of the findings. However, all referees raise a number of - often overlapping - issues that would need to be addressed before publication. All three referees indicate that further experiments are required to verify the localization of the pH-probes and the absence of mitopHlases in the IMS. Moreover, referee 2 and 3 both remark that additional experiments should address the role of apoptosis/STS.

Given these constructive comments, we would like to invite you to revise your manuscript with the understanding that the referee concerns (as detailed above and in their reports) must be fully addressed and their suggestions taken on board. Please address all referee concerns in a complete point-by-point response. Acceptance of the manuscript will depend on a positive outcome of a second round of review. It is EMBO reports policy to allow a single round of revision only and acceptance or rejection of the manuscript will therefore depend on the completeness of your responses included in the next, final version of the manuscript.

REFeree REPORTS

Referee #1:

The manuscript by Rosselin et al reported the role of L-OPA1 in mitoflash biogenesis, they have used array of genetically encoded pH-sensitive probes to monitor mitochondrial pH changes. Most of the experimental data are convincing and the study was excellently designed. Authors provide appropriate term "mitopHlashes" which is clearer to define mitochondrial pH-transients instead of mitoflashes. The utilization of genetically encoded pH-sensitive probes to identify the biogenesis of mitopHlashes is highly appreciated by the referee. Current study validated that mitopHlashes are indeed matrix pH-transients, not the cristae or intermembrane space pH change. Genetic deletion of Opa1 does not have any effect on $\Delta\psi$, however drastically reduce the efficiency of mitopHlash/ $\Delta\psi$ coupling. By using fusion-deficient OPA1K301A mutant, authors showed that mitochondrial fusion activity of OPA1 does not have any role in generation and regulation of mitopHlashes. The manuscript is clearly written and current study is of great interest for wide scientific community. However, there are some issues that need to be elaborated:

Specific Comments:

1. Authors showed mitochondrial localization of pH-probes using confocal microscopy (Figure 2, 3), indeed they are showing typical mitochondrial localization. However, authors should show their co-localization with already established mitochondrial markers to rule out any ambiguity.
2. Authors showed that OPA1 ablation did not affect the $\Delta\psi$, instead alters the mitoflash/ $\Delta\psi$ coupling. However, in their earlier study (Santo-Domingo et al 2013) they claim that OPA1 is necessary for mitoflash generation and "not a single pHmito flash was detected in Opa1-/- cells even after application of atractyloside". To contrast the earlier study from same group, in current study they found that "Using TMRM spikes as indicators of activity, they could then detect residual mitopHlashes events in Opa1-/- cells, indicating that Opa1 ablation severely impairs, but does not abrogate mitopHlash activity. Please elaborate this discrepancy.
3. Again their previous study claimed "Remarkably, Opa1 ablation abrogated pHmito flash activity (Figure 6C) and $\Delta\psi$ fluctuations (not shown)". Please clarify.
4. Authors claim that OPA1 stabilizes the respiratory complexes active conformation which enable respiring mitochondria to pH/ $\Delta\psi$ coupling. To validate this hypothesis authors should give biochemical experimental evidence for the involvement of L-OPA1 and fusion deficient OPA1 mutant in RCS assembly/disassembly.

Minor Comments:

1. Page 3, Line 10: Respiratory Chain Supercomplexes (RCS) instead of Respiratory Supercomplexes.
2. Description of Fig. 2G missing from the text.
3. Page 7: Although authors conducted an experiment using oligomycin and galactose rich media which does not have any effect on cristae or IMS pH, it is worth included as a supplementary information.
4. Description of Fig. 6E missing from the text.
5. Figure 2-7: Authors should show the images of TMRM for the depolarization (at least as a supplementary data).

Referee #2:

The authors present interesting data about mitochondrial flashes that contributes to the understanding of this phenomenon, which is currently not well understood. The data tells a compelling story about the role of matrix pH changes during mitochondrial flashes and how it is connected with mitochondrial membrane potential and requires OPA1 but not mitochondrial fusion. However, the IMS and cristae data is less convincing and would benefit from some additional

experimental evidence demonstrating that these probes work. The manuscript is well written and cites/discusses the literature. The manuscript could be improved with the following adjustments:

Major Concerns to be addressed:

1. Figure 2A and 3A: To show localization to mitochondria there should be a known mitochondrial marker. While the staining pattern of the images in Figures 2 and 3 appear to be mitochondrial, without comparing it to a known mitochondrial marker it cannot be definitively stated these images demonstrate mitochondrial localization. Therefore, it is recommended that either the wording in the text be changed as these experiments do not verify the localization or that the proper experiments are done with mitochondrial markers to demonstrate the localization (employing the Pearson's correlation coefficient to verify localization).
2. The cristae and IMS probe data suggest that there are too many potential technical issues to state that pH flashes are not detectable in the IMS and the cristae. Given that no treatment was able to demonstrate a change in IMS/cristae pH (including treatments that were expected to alter pH, such as Antimycin A) suggests that these probes are either not sensitive enough and/or not targeted properly, as the authors suggest. The authors should test additional compounds to demonstrate whether IMS or cristae pH changes can be detected with these probes, which would strengthen the conclusion that mitochondrial flashes do not occur in the IMS or cristae. Without additional evidence, I do not think the data supports the conclusion that mitopHlashes are matrix-limited.
3. Figure 5A: Representative images from a control cell should be added to demonstrate the normal diffusion of PA-GFP. Additionally, it does not appear that the same cell is always presented in the representative images at the 1 minute and 60 minute time points. Specifically for Opa1^{-/-} + WT OPA1, although this could be due to changes in morphology over the assay time. Please confirm that representative images are from the same cell.
4. Figure 7: It would be informative to know what STS does to the matrix pH, in addition to the depolarization rate and the coupling that is presented. Additionally, a 6 hour time point for the western blot analysis should be added to demonstrate cytosolic OPA1 is released with STS treatment. Quantification of the western blot and immunofluorescence should be included to verify the conclusions drawn from these experiments.

Minor Concerns/Comments:

1. Text and Figure legend/panels do not correlated for Figure 2 in a few places on page 6 of the manuscript (Figure 1F should be changes to 2F; 2F should be changed to 2G), please correct these issues.
2. Figure 3F: The y-axis title presents a ratio of F480/F380, while in Figure 3D the ratio is F488/F405 for the same probe. Confirm whether this is accurate.
3. Figure 5B: In addition to the increase in GFP area, it would be informative to analyze for the decrease in average fluorescence intensity over time. Additionally, please include how many cells were analyzed for each condition from the 3 independent experiments in the figure legend.
4. Page 10: Confirm whether the correct spelling is stauroporin (as written in the text) or stauroporine.
5. Specify pH/ m un/coupling instead of un/coupling alone so not to cause confusion that it could be related to mitochondrial respiration.

Referee #3:

The manuscript by Rosselin et al. is an interesting study that sheds further light onto the significance of mitochondrial flashes. In the present manuscript, the authors make the rather surprising discovery that mitoflashes do not occur in the IMS. They further demonstrate that apoptosis induction leads to

an uncoupling between mitoflashes and mitochondrial membrane potential changes. These are two rather important discoveries. The latter part of the paper is currently not well developed and it also appears not well connected to the remainder of the study. Overall, the study is of impressive quality and is based on powerful live cell imaging data. The detailed comments are found below.

1. The abstract is currently rather confusing. While this reviewer agrees with the renaming of mitoflashes, the abstract does not explain this appropriately and the reader is left confused about what are mitophlashes versus mitoflashes.
2. I would like to see some further controls on the different mitochondrial pH indicators. For instance, what happens with membrane permeabilization? Can digitonin be used? In Figure 2F, a ratiometric curve is shown, it should be translated into absolute pH units to get an idea of how much Antimycin A is able to alter the pH.
3. Although the authors have investigated this with matrix probes in their previous paper, an experiment with hyperfused mitochondria should be shown. Is the absence of flashes in the IMS uniform over larger areas? This absence of data is puzzling, since the authors show the Mfn1/2-/- control, but not this experiment.
4. Regarding the IMS pH sensors: how was their localization to the IMS verified? Simple IF does not tell much whether they have been correctly targeted. The discussion mentions that there was some mislocalization, but to which extent and how was it determined? Again, the antimycin A experiment lacks the pH units.
5. The STS experiment is interesting, but does not shed much light onto what is the originating cause of the uncoupling between pH and $\Delta\psi$ upon apoptosis induction. In their previous paper, the authors have shown that thapsigargin, another apoptosis inducer, did not alter flash frequency. How was this new condition based on STS different from thapsigargin? Would longer exposure with thapsigargin lead to an uncoupling? Is it really the commitment to the death program that is the determinant? A few other death triggers should be looked at to provide insight.
6. Generally speaking: Can the authors connect the STS experiment better to the remainder of the study? For instance, can the authors connect their observations to the Opa1 status, its oligomerization or its localization? The authors mention that coupling does not depend on Opa1 cleavage, but what happens to it upon apoptosis?

Minor points:

1. On page 6, Figures 1F and S1 are mentioned. Where are they?
2. The text regarding Figure 5 is confusing. The concept of Mfn1/2-/- is introduced, but not appropriately separated from the further Opa1 experiments.
3. The title of the results text describing Figure 6 is confusing. It states that long, uncleaved Opa1 mediates coupling, but in the paragraph, it is concluded that Opa1 processing is dispensable for mitoflash generation. Do the authors simply mean "Both OPA1 variants can mediate coupling"?

1st Revision - authors' response

29 October 2016

The comments of the reviewers are in *italics*, our responses to these comments are in plain letters.

Referee #1:

The manuscript by Rosselin et al reported the role of L-OPA1 in mitoflash biogenesis, they have used array of genetically encoded pH-sensitive probes to monitor mitochondrial pH changes. Most of the experimental data are convincing and the study was excellently designed. Authors provide appropriate term "mitopHlashes" which is clearer to define mitochondrial pH-transients instead of mitoflashes. The utilization of genetically encoded pH-sensitive probes to identify the biogenesis of

mitopHlashes is highly appreciated by the referee. Current study validated that mitopHlashes are indeed matrix pH-transients, not the cristae or intermembrane space pH change. Genetic deletion of Opa1 does not have any effect on $\Delta\Psi_m$, however drastically reduce the efficiency of mitopHlash/ $\Delta\Psi_m$ coupling. By using fusion-deficient OPA1K301A mutant, authors showed that mitochondrial fusion activity of OPA1 does not have any role in generation and regulation of mitopHlashes. The manuscript is clearly written and current study is of great interest for wide scientific community. However, there are some issues that need to be elaborated:

We thank the reviewer for the constructive comments. We have addressed the issues raised experimentally, and the new data strengthen our initial findings on the role of OPA1 in mitopHlash biogenesis. A point-by-point answer to the reviewer queries is provided below.

Specific Comments:

1. Authors showed mitochondrial localization of pH-probes using confocal microscopy (Figure 2, 3), indeed they are showing typical mitochondrial localization. However, authors should show their co-localization with already established mitochondrial markers to rule out any ambiguity.

As requested we have confirmed the mitochondrial localization of the probes using antibodies against-Hsp60 as an endogenous marker. As shown in supplementary figure EV1, all the probes co-localized with Hsp60 with Pearson and Mander's coefficients around 80%, thereby establishing their mitochondrial localization. In addition, we have performed immuno-EM with anti-GFP antibodies to precisely assess the localisation of our GFP-based pH probes in different mitochondrial compartments. These new data show that CIV8-spHlu is preferentially detected in cristae while MPP-spHlu is mainly detected in the matrix, as predicted from their targeting sequences. These data firmly establish the correct localization of the probes and are now included as Figure 3 (new).

2. Authors showed that OPA1 ablation did not affect the $\Delta\Psi_m$, instead alters the mitoflash/ $\Delta\Psi_m$ coupling. However, in their earlier study (Santo-Domingo et al 2013) they claim that OPA1 is necessary for mitoflash generation and "not a single pHmito flash was detected in Opa1-/- cells even after application of atractyloside". To contrast the earlier study from same group, in current study they found that "Using TMRM spikes as indicators of activity, they could then detect residual mitopHlashes events in Opa1-/- cells, indicating that Opa1 ablation severely impairs, but does not abrogate mitopHlash activity. Please elaborate this discrepancy.

Discussed together with point 3 below.

3. Again, their previous study claimed, "Remarkably, Opa1 ablation abrogated pHmito flash activity (Figure 6C) and $\Delta\Psi_m$ fluctuations (not shown)." Please clarify.

The detection of transient depolarization and mitopHlashes is challenging in Opa1^{-/-} cells whose mitochondria are totally fragmented and mobile. Imaging these stochastic transients requires a microscope with a good signal-to-noise ratio and an excellent resolution. The discrepancy between the two studies likely stems from the different sensitivity of the microscopes used to record mitoflashes. In our earlier publication, the experiments were performed on a spinning disk microscope while we used here a Nikon A1r laser scanning confocal microscope. The increased sensitivity of the newer microscope facilitated the visualization of both the TMRM and SypHer events, revealing the residual activity of Opa1^{-/-} cells. A representative experiment illustrating the challenge posed by the detection of sporadic depolarization transients occurring randomly in time and space within interconnected mitochondria is now shown in Fig EV5 and movie EV3. Science is a self-correcting endeavour and while we apologize for the confusion that the discrepancy with our earlier study might create, we believe that our newer and more accurate findings in fact strengthen and refine the conclusion that OPA1 plays an important role in mitopHlashes biogenesis. In support for our new findings, a recent study also reported residual mitopHlashes in OPA1-null cells, the low frequency of these residuals increasing upon osmotic challenge [1].

4. Authors claim that OPA1 stabilizes the respiratory complexes active conformation, which enables respiring mitochondria to pH/ $\Delta\Psi_m$ coupling. To validate this hypothesis authors should give biochemical experimental evidence for the involvement of L-OPA1 and fusion deficient OPA1 mutant in RCS assembly/disassembly.

We have attempted to quantify the formation of supercomplexes in the different cell lines on blue native gels, but despite our efforts failed to obtain reproducible results. These experiments are notoriously difficult and although we might be able to obtain valid data by refining the experimental conditions, we believe that the expected outcome of these biochemical experiments does not warrant the time, effort, and financial investment required. Instead, we have softened our conclusion and now only present supercomplex stabilization as one possible mechanism of the OPA1 action.

Minor Comments:

1. Page 3, Line 10: *Respiratory Chain Supercomplexes (RCS) instead of Respiratory Supercomplexes.*

Corrected

2. *Description of Fig. 2G missing from the text.*

Added

3. *Page 7: Although authors conducted an experiment using oligomycin and galactose rich media which does not have any effect on cristae or IMS pH, it is worth included as a supplementary information.*

We now include these experiments as Fig EV3. No pH change was detected with the pHluorin probes when oligomycin was added to the imaging media. A representative experiment conducted with Smac-rpHluorin is shown (Fig EV3E). When HeLa cells were grown in galactose media for 48h, no mitoflash was detected during membrane depolarization using the probes smac-rpHluorin, CVE-spHluorin and CIV8-spHluorin (Fig EV3G, representative experiment using CIV8-spHluorin). However a significant change in resting pH was observed compared to low glucose media (Fig EV3F), consistent with an earlier study [2].

4. *Description of Fig. 6E missing from the text.*

Added

5. *Figure 2-7: Authors should show the images of TMRM for the depolarization (at least as a supplementary data).*

Added (Fig EV5 and movies EV2 and EV3)

Referee #2:

The authors present interesting data about mitochondrial flashes that contributes to the understanding of this phenomenon, which is currently not well understood. The data tells a compelling story about the role of matrix pH changes during mitochondrial flashes and how it is connected with mitochondrial membrane potential and requires OPA1 but not mitochondrial fusion. However, the IMS and cristae data is less convincing and would benefit from some additional experimental evidence demonstrating that these probes work. The manuscript is well written and cites/discusses the literature. The manuscript could be improved with the following adjustments:

We thank the reviewer for the constructive comments. We have addressed the issues raised experimentally, and the new data confirm the correct localization of the IMS and cristae probes. A point-by-point answer to the reviewer queries is provided below.

Major Concerns to be addressed:

1. *Figure 2A and 3A: To show localization to mitochondria there should be a known mitochondrial marker. While the staining pattern of the images in Figures 2 and 3 appear to be mitochondrial, without comparing it to a known mitochondrial marker it cannot be definitively stated these images*

demonstrate mitochondrial localization. Therefore, it is recommended that either the wording in the text be changed as these experiments do not verify the localization or that the proper experiments are done with mitochondrial markers to demonstrate the localization (employing the Pearson's correlation coefficient to verify localization).

We have confirmed the mitochondrial localization of the probes using antibodies against-Hsp60 as an endogenous marker. All the probes co-localized with Hsp60 with Pearson and Mander's coefficients around 80% (Fig EV1), thereby establishing their mitochondrial localization. We have also performed immuno-EM with anti-GFP antibodies to precisely assess the localisation of our GFP-based pH probes in different mitochondrial compartments. These new data show that CIV8-spHlu is preferentially detected in cristae while MPP-spHlu is mainly detected in the matrix, as predicted from their targeting sequences. These data firmly establish the correct localization of the probes and are now included as Figure 3 (new).

2. The cristae and IMS probe data suggest that there are too many potential technical issues to state that pH flashes are not detectable in the IMS and the cristae. Given that no treatment was able to demonstrate a change in IMS/cristae pH (including treatments that were expected to alter pH, such as Antimycin A) suggests that these probes are either not sensitive enough and/or not targeted properly, as the authors suggest. The authors should test additional compounds to demonstrate whether IMS or cristae pH changes can be detected with these probes, which would strengthen the conclusion that mitochondrial flashes do not occur in the IMS or cristae. Without additional evidence, I do not think the data supports the conclusion that mitopHlashes are matrix-limited.

We now show that the cristae probe Cve-spHluorin reports different pH values when HeLa cells are grown in culture media permissive for oxidative phosphorylation or for glycolysis (Fig EV3F). Consistent with a previous study [2], we measured a resting pH of 7.2 in cells grown in 10 mM galactose and 7.0 in 5.5 mM glucose, an acidification that likely reflects the reverse activity of the ATP synthase in 5.5 mM glucose [3]. Together with our immuno-EM data (Fig 3) and the response of IMS and cristae probes to H⁺ ionophores (Fig 3), this demonstrates that our probes targeted to the IMS and cristae report acute and chronic changes in the pH of intra-mitochondrial compartments. We have tested a battery of pharmacological tools and failed to detect significant changes in IMS and cristae pH during acute inhibition of respiratory complexes. Without time-resolved control recordings of pH changes in IMS and cristae during acute RCS inhibition, we cannot rule out that mitopHlashes might propagate to these compartments without causing a detectable signal, although this appears unlikely given the amplitude of the concomitant matrix pH response. We have rephrased the text of our manuscript accordingly.

3. Figure 5A: Representative images from a control cell should be added to demonstrate the normal diffusion of PA-GFP. Additionally, it does not appear that the same cell is always presented in the representative images at the 1 minute and 60 minute time points. Specifically for Opa1^{-/-} + WT OPA1, although this could be due to changes in morphology over the assay time. Please confirm that representative images are from the same cell.

We have added pictures of a WT MEF cell in Figure 5A. The images taken at 1 and 60 minutes are from the same cell for each condition and this is now specified in the figure legend and in the material and methods section.

4. Figure 7: It would be informative to know what STS does to the matrix pH, in addition to the depolarization rate and the coupling that is presented. Additionally, a 6-hour time point for the western blot analysis should be added to demonstrate cytosolic OPA1 is released with STS treatment. Quantification of the western blot and immunofluorescence should be included to verify the conclusions drawn from these experiments.

Good point, because the mitopHlash frequency is affected by changes in the matrix pH. We have verified that the matrix pH remained stable for up to 2h during STS treatment. The data are shown below for the reviewer perusal and the pH values indicated in the text (page 11, line 4).

Effect of STS on matrix pH. Matrix pH was measured as described in material and methods in MEF cells expressing mito-sypHer and exposed to DMSO or 1 μ M STS for 2 h. Images were acquired on a Nikon A1r inverted confocal microscope. Values are mean \pm SD of 3 independent experiments.

We now include an 8h time point demonstrating that OPA1 is released in the cytosol following STS treatment (Fig 7C) and have quantified the extent of OPA1 and cytochrome c release (Fig 7C and 7D).

Minor Concerns/Comments:

1. Text and Figure legend/panels do not correlate for Figure 2 in a few places on page 6 of the manuscript (Figure 1F should be changes to 2F; 2F should be changed to 2G), please correct these issues.

Corrected, thank you.

2. Figure 3F: The y-axis title presents a ratio of F480/F380, while in Figure 3D the ratio is F488/F405 for the same probe. Confirm whether this is accurate.

The values are correct. A confocal microscope was used for the pH calibration shown in Fig 3D and a wide-field microscope for the time-resolved recordings shown in Fig 3F, using UV illumination matching the excitation peak of the probe.

3. Figure 5B: In addition to the increase in GFP area, it would be informative to analyze for the decrease in average fluorescence intensity over time. Additionally, please include how many cells were analyzed for each condition from the 3 independent experiments in the figure legend.

We have measured the decrease in PA-GFP fluorescence over time. These data are shown below for the reviewer perusal. The loss of PA-GFP fluorescence was more pronounced in WT and Opa1 rescued KO cells after 60 min, but the differences are not significant and we did not include these data in the revised MS. The number of cells analysed for each condition is now included in the figure legend.

Residual PA-GFP fluorescence intensity measured 60 min after photoactivation. PA-GFP fluorescence intensity was imaged in WT and Opa1^{-/-} cells transfected with the indicated plasmids 1 and 60 min following photoactivation and the extent of PA-GFP decrease calculated as $(F_{1min} - F_{60min})/F_{1min} * 100$. Data are mean \pm SD of 3 independent experiments.

4. Page 10: Confirm whether the correct spelling is stauroporin (as written in the text) or stauroporine.

Corrected (staurosporine).

5. Specify $pH/\Delta\Psi_m$ un/coupling instead of un/coupling alone so not to cause confusion that it could be related to mitochondrial respiration.

Corrected.

Referee #3:

The manuscript by Rosselin et al. is an interesting study that sheds further light onto the significance of mitochondrial flashes. In the present manuscript, the authors make the rather surprising discovery that mitoflashes do not occur in the IMS. They further demonstrate that apoptosis induction leads to an uncoupling between mitoflashes and mitochondrial membrane potential changes. These are two rather important discoveries. The latter part of the paper is currently not well developed and it also appears not well connected to the remainder of the study. Overall, the study is of impressive quality and is based on powerful live cell imaging data. The detailed comments are found below.

We thank the reviewer for the constructive comments and provide a point by point answer to the detailed comments below.

1. *The abstract is currently rather confusing. While this reviewer agrees with the renaming of mitoflashes, the abstract does not explain this appropriately and the reader is left confused about what are mitophlashes versus mitoflashes.*

We now explain the new terminology in the abstract.

2. *I would like to see some further controls on the different mitochondrial pH indicators. For instance, what happens with membrane permeabilization? Can digitonin be used? In Figure 2F, a ratiometric curve is shown, it should be translated into absolute pH units to get an idea of how much Antimycin A is able to alter the pH.*

We now provide recordings translated into absolute pH values to illustrate the effects of Antimycin A in Fig 2F and Fig EV3. We also show that the cristae probe Cve-spHluorin reports different pH values when HeLa cells are grown in culture media permissive for oxidative phosphorylation or for glycolysis (Fig EV3F). Consistent with a previous study [2], we measured a resting pH of 7.2 in cells grown in 10 mM galactose and 7.0 in 5.5 mM glucose, an acidification that likely reflects the reverse activity of the ATP synthase in 5.5 mM glucose [3]. Together with our immuno-EM data

(Fig 3) and the response of IMS and cristae probes to H⁺ ionophores (Fig 2), this demonstrates that our probes targeted to the IMS and cristae report acute and chronic changes in the pH of intra-mitochondrial compartments. We had previously documented the effect of respiratory chain inhibitors on the matrix mitopHlashes [4]. None of these inhibitors evoked significant acute changes in IMS or cristae pH when applied (Fig. EV3C-E), and flashing events were not detected in the cristae of cells cultured in galactose-rich media (Fig. EV3G). As discussed in the response to reviewer 2, without time-resolved control recordings of pH changes in IMS and cristae during acute RCS inhibition, we cannot rule out that mitopHlashes might propagate to these compartments without causing a detectable signal, although this appears unlikely given the amplitude of the concomitant matrix pH response. The effects of acute addition of 100 μ M digitonin are shown below:

Upon digitonin addition, mitochondria fragmented and the fluorescence of the matrix and cristae probes became sensitive to imposed changes in the pH of the extracellular solution. This is consistent with our earlier observation that cytosolic pH changes rapidly propagate to the mitochondrial matrix [5].

3. Although the authors have investigated this with matrix probes in their previous paper, an experiment with hyperfused mitochondria should be shown. Is the absence of flashes in the IMS uniform over larger areas? This absence of data is puzzling, since the authors show the Mfn1/2-/- control, but not this experiment.

As suggested, we have measured pH changes in the IMS and cristae of cells expressing a dominant-negative DRP1 to enforce mitochondrial fusion. The mitochondrial area exhibiting stochastic drops in DPM measured with TMRM was greatly increased in cells with hyperfused mitochondria (Fig EV4), but pH flashes remained undetectable with smac-rpHluorin, CIV8-spHluorin and CVe-pHluorin. We also used hyperosmotic stress conditions known to increase flash frequency [6] but failed to detect pH flashes with the probes targeted to the cristae or the IMS (data not shown).

4. Regarding the IMS pH sensors: how was their localization to the IMS verified? Simple IF does not tell much whether they have been correctly targeted. The discussion mentions that there was some mislocalization, but to which extent and how was it determined? Again, the antimycin A experiment lacks the pH units.

We now show that all our probes co-localize with the mitochondrial marker Hsp60 (Fig EV2) and have quantified the distribution of a cristae and matrix probe in mitochondrial subcompartments by immunogold electron microscopy (Fig 3, new). As expected, CIV8-spHluorin immunoreactivity was mainly detected in the cristae and the IMS, the residual immunoreactivity detected in the matrix and

outside mitochondria reflecting either mislocalization or non-specific binding. The antimycin A recording is now reported in pH units.

5. *The STS experiment is interesting, but does not shed much light onto what is the originating cause of the uncoupling between pH and $\Delta\Psi$ upon apoptosis induction. In their previous paper, the authors have shown that thapsigargin, another apoptosis inducer, did not alter flash frequency. How was this new condition based on STS different from thapsigargin? Would longer exposure with thapsigargin lead to an uncoupling? Is it really the commitment to the death program that is the determinant? A few other death triggers should be looked at to provide insight.*

We previously used thapsigargin to study the effect of ER Ca^{2+} store depletion on mitopHlashes, a process developing on a time scale too short to induce detectable apoptosis (8 min vs. 2h). As suggested, we have analysed the effect of two other death triggers, H_2O_2 and etoposide. Our new data show that these two agents induce a significant decrease in pH/ $\Delta\Psi$ m coupling (Fig EV5A and EV5B). We thank the reviewer for suggesting this experiment.

6. *Generally speaking: Can the authors connect the STS experiment better to the remainder of the study? For instance, can the authors connect their observations to the Opa1 status, its oligomerization or its localization? The authors mention that coupling does not depend on Opa1 cleavage, but what happens to it upon apoptosis?*

We now show that pH/ Dy_m uncoupling is associated with the release of OPA1 in the cytosol (Fig 7C). The intensity of the OPA1 reactive band detected in the cytosolic fraction progressively increased after 2h and 8h of staurosporine exposure.

Minor points:

1. *On page 6, Figures 1F and S1 are mentioned. Where are they?*

Corrected, thank you.

2. *The text regarding Figure 5 is confusing. The concept of Mfn1/2^{-/-} is introduced, but not appropriately separated from the further Opa1 experiments.*

We have rephrased this section to better integrate the Mfn1/2^{-/-} experiment with the general context of our study.

3. *The title of the results text describing Figure 6 is confusing. It states that long, uncleaved Opa1 mediates coupling, but in the paragraph, it is concluded that Opa1 processing is dispensable for mitoflash generation. Do the authors simply mean "Both OPA1 variants can mediate coupling"?*

Fig. 6E shows that the proteolytic processing of OPA1 is not required for pH/ $\Delta\Psi$ m coupling. This implies that the long, uncleaved OPA1 forms are sufficient to mediate coupling since in the absence of the Oma1 and Yme1 proteases only the long forms of OPA1 remain. Fig. 6B shows that either the V1 or the V7 variant can reconstitute coupling in Opa1 null cells, implying that the two variants are redundant for this function. The logical conclusion is, as the reviewer suggests, that both OPA1 variants can mediate coupling, without requiring processing. We changed the title of Fig 6 accordingly.

References

1. Huang Z, Zhang W, Gong G, Fang H, Zheng M, Wang X, Xu J, Dirksen RT, Sheu S-S, Cheng H, *et al.* (2011) Response to "A critical evaluation of cpYFP as a probe for superoxide". *Free Radical Biology and Medicine* **51**: 1937-1940
2. Rieger B, Junge W, Busch KB (2014) Lateral pH gradient between OXPHOS complex IV and F(0)F(1) ATP-synthase in folded mitochondrial membranes. *Nature communications* **5**: 3103
3. Campanella M, Parker N, Tan CH, Hall AM, Duchen MR (2009) IF(1): setting the pace of the F(1)F(o)-ATP synthase. *Trends in biochemical sciences* **34**: 343-50

4. Santo-Domingo J, Giacomello M, Poburko D, Scorrano L, Demaurex N (2013) OPA1 promotes pH flashes that spread between contiguous mitochondria without matrix protein exchange. *The EMBO journal* **32**: 1927-40
5. Poburko D, Santo-Domingo J, Demaurex N (2011) Dynamic regulation of the mitochondrial proton gradient during cytosolic calcium elevations. *The Journal of biological chemistry* **286**: 11672-84
6. Abad MFC, Di Benedetto G, Magalhães PJ, Filippin L, Pozzan T (2004) Mitochondrial pH Monitored by a New Engineered Green Fluorescent Protein Mutant. *Journal of Biological Chemistry* **279**: 11521-11529

2nd Editorial Decision

12 December 2016

Thank you for the submission of your revised manuscript to EMBO reports. I apologize for the delay in getting back to you with a decision on your manuscript. We were still hoping to receive feedback from referee 2 on it, but despite serious efforts from our side, this reviewer has not submitted his/her review. Therefore, I will make a decision based on the two reports we have received so far and which are both positive. Moreover, both referees indicated that the concerns of referee 2 have been adequately addressed in their opinion.

As you will see, referee 3 suggests some minor changes that need to be addressed before we can accept your manuscript for publication. From the editorial side, there are also a few things that we need before we can proceed with the official acceptance of your study.

REFEREE REPORTS

Referee #1:

The results and the data in the revised manuscript has been significantly improved. All the reviewer's concerns were thoroughly addressed. It is worth publishing in the EMBO Reports.

Referee #3:

The authors have addressed all comments by myself and the other reviewers adequately. The manuscript is now of very high technical quality. This is a very nice work that will be well respected within the community. Very few minor changes or suggestions should be considered:

1. The contemporaneous nature of pH and changes should be shown in an aligned, zoomed-in way in Figure 1H, where we can see whether the two are perfectly aligned or whether there is maybe a small time difference between the two readouts. Were any differences in the exact time points of rises/falls of the two measurements EVER found?
2. The graph in Figure 4C appears to not be representative, when compared to 4E.
2. Typo on page 4, top: "transient", not "transients". Typo on page 9, bottom: "Mfn", not "Mnf".

2nd Revision - authors' response

19 December 2016

Referee #1:

The results and the data in the revised manuscript has been significantly improved. All the reviewer's concerns were thoroughly addressed. It is worth publishing in the EMBO Reports.

We thank the reviewer for the positive comments.

Referee #3:

The authors have addressed all comments by myself and the other reviewers adequately. The manuscript is now of very high technical quality. This is a very nice work that will be well respected within the community. Very few minor changes or suggestions should be considered:

1. The contemporaneous nature of pH and changes should be shown in an aligned, zoomed-in way in Figure 1H, where we can see whether the two are perfectly aligned or whether there is maybe a small time difference between the two readouts. Were any differences in the exact time points of rises/falls of the two measurements EVER found?

Figure 1H was modified. As we previously reported (Santo-Domingo et al., EMBO J, 2013) the upstroke of a mitopHlash is always perfectly coincidental with the onset of a depolarisation event and we could not detect any lag in the propagation of one of the two components of the signal even at the highest resolution possible in our confocal microscope (2 ms in line scan mode). On the other hand, the decay phase of a mitopHlash is monotonic while the mitochondrial potential can exhibit delayed recovery. Since we previously illustrated this behaviour in our previous publication (Fig 4C of Santo-Domingo et al., EMBO J, 2013), we feel that it would be redundant to illustrate again that the changes in pH and do not always mirror each other during the recovery phases but discuss this point in the text (p. 7).

2. The graph in Figure 4C appears to not be representative, when compared to 4E.

The discrepancy between the illustrative recordings in Fig 4C and the aggregate data in Fig 4E reflects the "all or none" behaviour of Opa1 null cells re-expressing OPA1. These cells either exhibited mitopHlashes with every depolarization event (100% coupling as illustrated in Fig 4C) or no mitopHlash associated with depolarization events (0% coupling, as shown in Fig 4B), resulting in an average coupling rate of 50%. We now discuss this point but prefer to keep the original recording to illustrate ability of OPA1 re-expression to restore a normal phenotype.

2. Typo on page 4, top: "transient", not "transients." Typo on page 9, bottom: "Mfn", not "Mnf".

Corrected

3rd Editorial Decision

21 December 2016

I am very pleased to accept your manuscript for publication in the next available issue of EMBO reports. Thank you for your contribution to our journal.

Corresponding Author Name: Nicolas Demaurex

Manuscript Number: EMBOR-2016-42931V1